# COVARIATE-MODERATED EMPIRICAL BAYES MATRIX FACTORIZATION

## ABSTRACT

Matrix factorization is a fundamental method in statistics and machine learning for inferring and summarizing structure in multivariate data. Modern data sets often come with "side information" of various forms (images, text, graphs) that can be leveraged to improve estimation of the underlying structure. However, existing methods that leverage side information are limited in the types of data they can incorporate, and they assume specific parametric models. Here, we introduce a novel method for this problem, *covariate-moderated empirical Bayes matrix factorization* (cEBMF). cEBMF is a modular framework that accepts any type of side information that is processable by a probabilistic model or neural network. The cEBMF framework can accommodate different assumptions and constraints on the factors through the use of different priors, and it adapts these priors to the data. We demonstrate the benefits of cEBMF in simulations and in analyses of spatial transcriptomics and MovieLens data.

## 1 INTRODUCTION

Matrix factorization methods, which include principal component analysis (PCA), factor analysis, and non-negative matrix factorization (NMF) (Lee and Seung, 1999), are very widely used methods for inferring latent structure from data, performing exploratory data analyses and visualizing large data sets (e.g., Alexander et al. 2023; Novembre and Stephens 2008; Sainburg et al. 2020). Matrix factorization methods are also instrumental in other statistical analyses such as adjusting for unobserved confounding (Leek and Storey, 2007). Recent innovations in matrix factorization methods include the development of sparse matrix factorizations such as sparse PCA (Zou et al., 2006) and sparse factor analysis (Engelhardt and Stephens, 2010; Lan et al., 2014). In these works, sparsity is generally induced through a penalty or a prior distribution; however, as in many penalized regression problems (e.g., Tibshirani 1996), selecting the "right" amount of regularization, or the "best" prior, is an unresolved question, or may be reliant on cross-validation techniques that are inconvenient and computationally burdensome

When factorizing a matrix, say $\mathbf{Z}$, the matrix may be accompanied with additional row or column data—"side information"—that may be able to "guide" the matrix factorization algorithm toward a more accurate or interpretable factorization. A recent prominent example of this in genomics research is spatial transcriptomics data (Marx, 2021), which is expression profiled in many genes at many spatial locations ("pixels") (Vandereyken et al., 2023). For a variety of reasons, one typically seeks to factorize $\mathbf{Z}$, which is the matrix of gene expression profiles, but the 2-d coordinates of the pixels also provide important information about the biological context of the cells; for example, we might expect nearby pixels to belong to the same cell type or tissue region. Therefore, "spatially aware" matrix factorization methods have recently been proposed for spatial transcriptomics data (Shang and Zhou, 2022; Townes and Engelhardt, 2023; Velten et al., 2022).

In this paper, we describe a novel matrix factorization framework that allows high-dimensional row and column data to guide the factorizations without having to make specific assumptions about how these data inform the factorization. For example, although our framework can be applied to data that exhibit spatial properties, it does not assume or require that the data be spatial. (Further, there are sometimes benefits to not making strong assumptions about the spatial organization of the data even when we know the data are spatial.) Our framework is also flexible in that it includes many existing approaches, including unconstrained matrix factorization (Wang and Stephens, 2021;

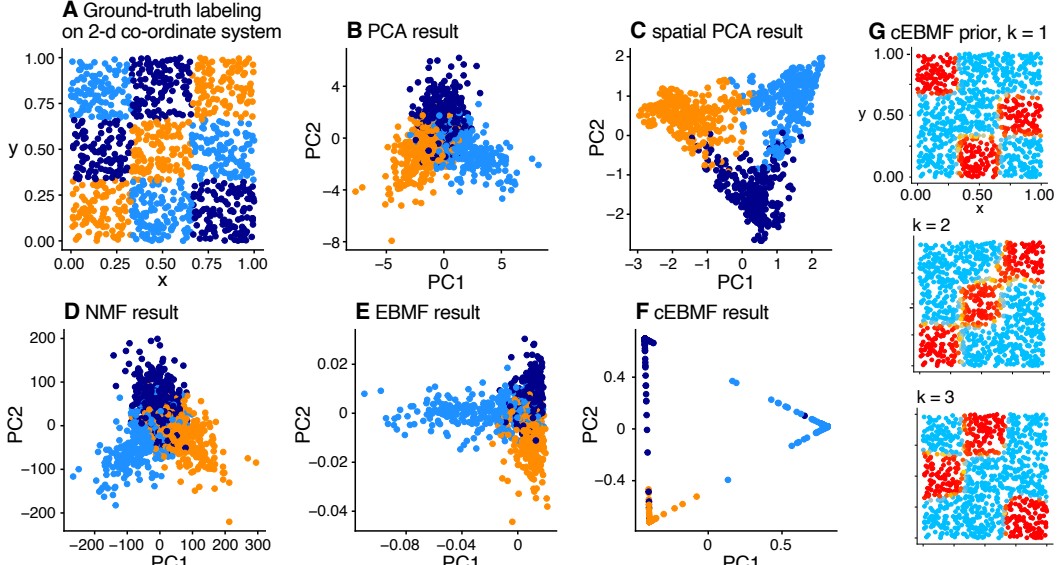

Figure 1: Toy simulation illustrating cEBMF for learning a matrix factorization, $\mathbf{Z} \approx \mathbf{L}\mathbf{F}^T$. In this example, $\mathbf{Z}$ is a $1,000 \times 200$ matrix, and each of the $N = 1,000$ data points is assigned to one of three clusters (orange, light blue, dark blue). Points near each other tend to be assigned to the same cluster, except near boundaries (A). Without the side information (the 2-d coordinates in A), PCA, NMF and EBMF with $K = 3$ factors cluster some points accurately, but not others (B, D, E). By contrast, Spatial PCA (Shang and Zhou, 2022) and cEBMF, by incorporating the side information into the prior, more accurately cluster the points (C, F). (For visualization purposes, the $\mathbf{L}$ matrix from NMF, EBMF and cEBMF is projected onto the top 2 PCs in D, E, F.) Whereas Spatial PCA assumes the data points are spatial, cEBMF does not, and instead has a flexible prior that is adapted to the data; G visualizes this prior (specifically, the color of the points depicts $\pi_{ik} := \pi(\boldsymbol{x}_i, \boldsymbol{\theta}_k)$, $k = 1, 2, 3$, the prior probability that element $i$ in column $k$ of $\mathbf{L}$ is nonzero, red = high probability, blue = low probability). See Sections 3 and 4 for model definitions and details of the simulation.

Zhong et al., 2022), non-negative matrix factorization (Lin and Boutros, 2020), semi-non-negative matrix factorization (Ding et al., 2010), and more recent methods that incorporate side information Wang et al. (2024), all as special cases.. These features are achieved by taking an empirical Bayes approach, building on recent work on empirical Bayes matrix factorization (EBMF) (Wang and Stephens, 2021; Zhong et al., 2022). In particular, we extend the EBMF approach of Wang and Stephens (2021) to allow for adaptive priors that are modified by the side information. We call this approach "covariate-moderated empirical Bayes matrix factorization", or "cEBMF" for short. See Fig. 1 for a toy example that illustrates the key features of cEBMF.

## 2 RELATED WORK

The literature on matrix factorization methods that incorporate side information is quite extensive. The different methods make different modeling assumptions, and are typically motivated by certain types of data. Although it is infeasible to review all relevant literature here, we mention a few of the more important methods, and highlight the methods that are most closely related to cEBMF.

Several variants of the topic model—which can be viewed as a matrix factorizations with "sum-to-one" constraints on $\mathbf{L}$ and $\mathbf{F}$ (Carbonetto et al., 2021)—incorporate side information in different ways; for example, the correlated topic model (Lafferty and Blei, 2005) and the structural topic model (Roberts et al., 2016) incorporate document-level side information into the priors on $\mathbf{L}$.

Collective matrix factorization (CMF) (Bouchard et al., 2013; Lee and Choi, 2009; Singh and Gordon, 2008) has gained considerable interest, but is based on ideas that are quite different from cEBMF: like cEBMF, CMF assumes that the side information is the form of a matrix; but unlike

cEBMF, CMF assumes that the side information factorizes in a similar way to $\mathbf{Z}$. Clearly, the CMF assumpion will not make sense for some applications.

Another prominent theme in matrix factorization with side information is incorporating group-level or categorical side information (including ontologies). Among the several methods in this area include CTPF Gopalan et al. (2014) and the method of Hu et al. (2016).

The method most closely related to cEBMF is MFAI (Wang et al., 2024) (see also Porteous et al. 2010 for related ideas). MFAI is in fact a special case of cEBMF in which the priors on $\mathbf{F}$ are normal and the prior means are informed by the covariates. Similar to cEBMF, MFAI allows these priors to be adapted separately for each factor $k$. However, it is not nearly as general as cEBMF; it implements only a single model, a single prior family of a specific parametric form, a specific procedure for fitting these priors (using gradient boosted tree methods; Friedman 2001), and it only accommodates row-wise side information.

For spatial transcriptomics data (which we consider in Sec. 4.3), Spatial PCA (Shang and Zhou, 2022) models the spatial similarity among the rows of $\mathbf{Z}$ using Gaussian process priors. (Spatial PCA is closely related to GP-LVM; Lawrence and Moore 2007, see also Zhou et al. 2012.) An NMF version of this approach generates "parts-based representations" guided by the spatial context of the data points (Townes and Engelhardt, 2023). IRIS (Ma and Zhou, 2024), regularizes the factors through a penalty function that encodes the spatial information in a graph (see also Cai et al. 2011).

## 3 COVARIATE-MODERATED EMPIRICAL BAYES MATRIX FACTORIZATION

### 3.1 BACKGROUND: EMPIRICAL BAYES MATRIX FACTORIZATION

Empirical Bayes matrix factorization (EBMF) (Wang and Stephens, 2021; Willwersheid, 2021) is a flexible framework for matrix factorization: it approximates a matrix $\mathbf{Z} \in \mathbb{R}^{n \times p}$ as the product of two low-rank matrices,

$$\mathbf{Z} \approx \mathbf{L}\mathbf{F}^T, \tag{1}$$

where $\mathbf{L} \in \mathbb{R}^{n \times K}$, $\mathbf{F} \in \mathbb{R}^{p \times K}$, and $K \geq 1$. (Typically, $K \ll n, p$.) EBMF assumes a normal model of the data,

$$\mathbf{Z} = \mathbf{L}\mathbf{F}^T + \mathbf{E}, \quad e_{ij} \sim \mathcal{N}(0, \tau_{ij}^{-1}), \tag{2}$$

in which $\mathcal{N}(\mu, \sigma^2)$ denotes the normal distribution with mean $\mu$ and variance $\sigma^2$, and the residual variances $\tau_{ij}^{-1}$ may vary by row ($i$) or column ($j$) or both. (EBMF, and by extention cEBMF, also allows $\mathbf{Z}$ to contain missing values, which is important for some applications of matrix factorization, including collaborative filtering; see Sec. 4.2.) EBMF assumes prior distributions for elements of $\mathbf{L}$ and $\mathbf{F}$, which are themselves estimated among prespecified prior families $\mathcal{G}_{\ell,k}$ and $\mathcal{G}_{f,k}$:

$$\begin{aligned}
\ell_{ik} \sim g_k^{(\ell)}, \quad g_k^{(\ell)} \in \mathcal{G}_{\ell,k}, \quad k = 1, \ldots, K \\
f_{jk} \sim g_k^{(f)}, \quad g_k^{(f)} \in \mathcal{G}_{f,k}, \quad k = 1, \ldots, K.
\end{aligned} \tag{3}$$

The flexibility of EBMF comes from the wide range of different possible prior families (including non-parametric families). Different choices of prior family correspond to different existing matrix factorization methods. For example, if all families $\mathcal{G}_{\ell,k}$ and $\mathcal{G}_{\ell,k}$ are the family of zero-mean normal priors, then $\mathbf{L}\mathbf{F}^T$ is similar to a truncated singular value decomposition (SVD) (Nakajima and Sugiyama, 2011). When the prior families are all point-normal (mixtures of a point mass at zero and a zero-centered normal), one obtains empirical Bayes versions of sparse SVD or sparse factor analysis (Engelhardt and Stephens, 2010; Witten et al., 2009; Yang et al., 2014). The prior families can also constrain $\mathbf{L}$ and $\mathbf{F}$; for example, families that only contain distributions with non-negative support result in empirical Bayes versions of non-negative matrix factorization (NMF) (Gillis, 2021; Lee and Seung, 1999). In summary, EBMF (2–3) is a highly flexible modeling framework for matrix factorization that includes important previous methods as special cases, but also many new combinations (e.g., Liu et al. 2023).

### 3.2 THE CEBMF MODEL

In covariate-moderated EBMF (cEBMF), we assume that we have some "side information" (covariates) for rows and/or columns of $\mathbf{Z}$ (Adams et al., 2010; Virshup et al., 2024). Let $\boldsymbol{x}_i$ denote the

available information for the $i$-th row of $\mathbf{Z}$, and let $\boldsymbol{y}_j$ denote the available information for the $j$-th column of $\mathbf{Z}$. In principle, $\boldsymbol{x}_i$ and $\boldsymbol{y}_j$ can be any information processable by a neural net (text, graph, image, or other structured data), but for simplicity we assume that this information is stored in matrix form. Let $\mathbf{X} \in \mathbb{R}^{n \times p_{\mathbf{X}}}$ be a matrix containing information on the rows of $\mathbf{Z}$, with $\boldsymbol{x}_i$ corresponding to the $i$-th row of $\mathbf{X}$ (e.g., $\boldsymbol{x}_i$ might contain the 2-d coordinate of the cell $i$). Similarly, let $\mathbf{Y} \in \mathbb{R}^{p \times p_{\mathbf{Y}}}$ contain information on the columns of $\mathbf{Z}$, with $\boldsymbol{y}_j$ corresponding to the $j$-th row of $\mathbf{Y}$. In cEBMF, we incorporate this side information into the model through parameterized priors:

$$
\begin{aligned}
\ell_{ik} &\sim g_k^{(\ell)}(\boldsymbol{x}_i), \quad g_k^{(\ell)}(\boldsymbol{x}_i) \in \mathcal{G}_{\ell,k}, \quad k = 1, \dots, K \\
f_{jk} &\sim g_k^{(f)}(\boldsymbol{y}_j), \quad g_k^{(f)}(\boldsymbol{y}_i) \in \mathcal{G}_{f,k}, \quad k = 1, \dots, K,
\end{aligned}
\tag{4}
$$

where $g_k^{(\ell)}(\boldsymbol{x}_i)$ is a probability distribution within the family $\mathcal{G}_{\ell,k}$, parameterized by $\boldsymbol{x}_i$ and $g_k^{(f)}(\boldsymbol{y}_j)$ is a probability distribution within the family $\mathcal{G}_{f,k}$ parameterized by $\boldsymbol{y}_i$.

A key limitation of many existing approaches is that they integrate the side information using restrictive parametric models. Additionally, the chosen priors may make strong or perhaps unrealistic assumptions about the structure underlying the data; for example, Gaussian process priors, which have been used in this setting (e.g., Lawrence 2005; Lawrence and Moore 2007; Shang and Zhou 2022), typically assume that the factors vary smoothly in space, which makes it difficult to accurately capture sharp changes at boundaries (Hyoung-Moon Kim and Holmes, 2005). Existing methods also usually rely on hyperparameters that need to be tuned or selected (e.g., using cross-validation).

To address these issues, we propose cEBMF, a method that:

1. Can leverage a large variety of models (e.g., multinomial regression, multilayer perceptron, graphical neural nets, convolutional neural nets) to integrate the side information into the prior.

2. Can use families of priors that are flexible in form and thus do not make strong assumptions.

3. Allows automatic selection of the hyperparameters in (3) using an empirical Bayes approach.

More formally, we fit a model for each column $k$ of $\mathbf{L}$ (and similarly for each column $k$ of $\mathbf{F}$) that maps each vector of covariates $\boldsymbol{x}_i$ to a given element $g_k^{(\ell)} \in \mathcal{G}_{\ell,k}$. In Sec. 3.3 we describe a simple yet general algorithm that simultaneously learns the factors $\mathbf{L}, \mathbf{F}$ and priors $g_k^{(\ell)}(\boldsymbol{x}_i), g_k^{(f)}(\boldsymbol{y}_j)$.

### 3.2.1 AN ILLUSTRATION: CEBMF WITH SIDE INFORMATION ON FACTOR SPARSITY

Here we illustrate the implementation of the cEBMF framework using a prior family that is simple yet broadly applicable. This prior family assumes that the covariates $\mathbf{X}, \mathbf{Y}$ *only inform the pattern of sparsity*—that is, the placement of zeros—in $\mathbf{L}, \mathbf{F}$. This type of prior is of particular interest in the matrix factorization setting because matrix factorizations typically have the hidden complication that they are invariant to rescaling; therefore, priors that inform the magnitude of $\ell_{ik}, f_{jk}$ are difficult to design. (By "invariant to rescaling," we mean that the likelihood or objective does not change if we replace $\mathbf{L}\mathbf{F}^T$ by $\tilde{\mathbf{L}}\tilde{\mathbf{F}}^T$, where $\tilde{\mathbf{L}} = \mathbf{L}\mathbf{D}, \tilde{\mathbf{F}} = \mathbf{F}\mathbf{D}^{-1}$, and $\mathbf{D}$ is an invertible diagonal matrix.)

We define the prior family $\mathcal{G}_{\text{ss}}$ as

$$
\mathcal{G}_{\text{ss}} := \{g : g(u) = (1 - \pi(\boldsymbol{x}, \boldsymbol{\theta}))\delta_0(u) + \pi(\boldsymbol{x}, \boldsymbol{\theta})g_1(u; \boldsymbol{\omega})\},
\tag{5}
$$

in which $\delta_0(u)$ denotes the point-mass at zero, $g_1(u, \boldsymbol{\omega})$ denotes the density of a probability distribution $g_1(\boldsymbol{\omega})$ on $u \in \mathbb{R}$, and $\boldsymbol{x} \in \mathbb{R}^m$ denotes the covariate. When $g_1$ is the normal distribution, (5) is the family of "spike-and-slab" priors (Mitchell and Beauchamp, 1988), and cEBMF with $\mathcal{G}_{\ell,k} = \mathcal{G}_{\text{ss}}, \mathcal{G}_{f,k} = \mathcal{G}_{\text{ss}}$ implements a version of sparse factor analysis (Engelhardt and Stephens, 2010; Witten et al., 2009; Yang et al., 2014) in which the sparsity of the factors is informed by the covariates. (The "ss" subscript in $\mathcal{G}_{\text{ss}}$ is short for "spike-and-slab.") Alternatively, if $g_1$ is a distribution with support on $u \in \mathbb{R}_+$, such as an exponential distribution, then cEBMF implements a version of sparse NMF.

The free parameters are $\boldsymbol{\theta}$, which controls the weight on the "spike" $\delta_0$, and $\boldsymbol{\omega}$, which controls the shape of the "slab" $g_1$. A straightforward parameterization is with a logistic regression model,

$$
\pi(\boldsymbol{x}, \boldsymbol{\theta}) = \phi\big(\theta_0 + \sum_{t=1}^{T} x_t \theta_t\big),
\tag{6}
$$

where $\phi(x) := 1/(1 + e^{-x})$ denotes the sigmoid function, and $\boldsymbol{\theta} \in \mathbb{R}^{T+1}$.

### 3.3 THE cEBMF LEARNING ALGORITHM

A key feature of the cEBMF modeling framework is that the algorithm for fitting the priors and estimating the factors is simple to describe and often straightforward to implement. In brief, the cEBMF learning algorithm reduces to a series of *covariate-moderated empirical Bayes normal means (cEBNM) problems* (Willwerscheid et al., 2024) so that any methods that solves the cEBNM problem can be "plugged in" to the generic cEBMF algorithm. Thus, the cEBMF framework naturally lends itself to modular algorithm design and software implementation.

#### 3.3.1 BACKGROUND: EMPIRICAL BAYES NORMAL MEANS

Given $n$ observations $\hat{\beta}_i \in \mathbb{R}$ with known standard deviations $s_i > 0$, $i = 1, \dots, n$, the normal means model (Efron and Morris, 1972; Robbins, 1951; Stephens, 2017) is

$$\hat{\beta}_i \overset{\text{ind.}}{\sim} N(\beta_i, s_i^2), \tag{7}$$

where the "true" means $\beta_i \in \mathbb{R}$ are unknown. We further assume that

$$\beta_i \overset{\text{i.i.d.}}{\sim} g \in \mathcal{G}, \tag{8}$$

where $\mathcal{G}$ is some prespecified family of probability distributions. The empirical Bayes (EB) approach to fitting this model exploits the fact that the noisy observations $\hat{\beta}_i$, contain not only information about the underlying means $\beta_i$ but also about how the means are collectively distributed (i.e., $g$). EB approaches "borrow information" across the observations to estimate $g$, typically by maximizing the marginal log-likelihood. The unknown means $\beta_i$ are generally estimated by their posterior mean.

To adapt EBNM (7–8) to the cEBMF framework, we allow the prior for the $i$-th unknown mean to depend on additional data $\boldsymbol{d}_i$,

$$\beta_i \overset{\text{ind.}}{\sim} g(\boldsymbol{d}_i, \boldsymbol{\theta}) \in \mathcal{G}, \tag{9}$$

so that each combination of $\boldsymbol{\theta}$ and $\boldsymbol{d}_i$ maps to an element of $\mathcal{G}$. We refer to this modified EBNM model as "covariate-moderated EBNM" (cEBNM).

Solving the cEBNM problem therefore involves two key computations:

**1. Estimate the model parameters.** Compute

$$\hat{\boldsymbol{\theta}} := \underset{\boldsymbol{\theta} \in \mathbf{R}^m}{\operatorname{argmax}} \, \mathcal{L}(\boldsymbol{\theta}), \tag{10}$$

where $\mathcal{L}(\boldsymbol{\theta})$ denotes the marginal likelihood,

$$\mathcal{L}(\boldsymbol{\theta}) := p(\hat{\boldsymbol{\beta}} \mid \boldsymbol{s}, \boldsymbol{\theta}, \mathbf{D}) = \prod_{i=1}^{n} \int \mathcal{N}(\hat{\beta}_i; \beta_i, s_i^2) \, g(\beta_i; \boldsymbol{d}_i, \boldsymbol{\theta}) \, d\beta_i, \tag{11}$$

in which $\hat{\boldsymbol{\beta}} = (\hat{\beta}_1, \dots, \hat{\beta}_n)$, $\boldsymbol{s} = (s_1, \dots, s_n)$, $\mathbf{D}$ is a matrix storing $\boldsymbol{d}_1, \dots, \boldsymbol{d}_n$, $\mathcal{N}(\hat{\beta}_i; \beta_i, s_i^2)$ denotes the density of $\mathcal{N}(\beta_i, s_i^2)$ at $\hat{\beta}_i$, and $g(\beta_i; \boldsymbol{d}_i, \boldsymbol{\theta})$ denotes the density of $g(\boldsymbol{d}_i, \boldsymbol{\theta})$ at $\beta_i$.

**2. Compute posterior summaries.** Compute summaries from the posterior distributions, such as the posterior means $\bar{\beta}_i := \mathbb{E}[\beta_i \mid \hat{\beta}_i, s_i, \hat{\boldsymbol{\theta}}, \mathbf{D}]$, using the estimated prior,

$$p(\beta_i \mid \hat{\beta}_i, s_i, \hat{\boldsymbol{\theta}}, \mathbf{D}) \propto \mathcal{N}(\hat{\beta}_i; \beta_i, s_i^2) \, g(\beta_i; \boldsymbol{d}_i, \hat{\boldsymbol{\theta}}). \tag{12}$$

For many classical prior families, such as the prior family in Sec. 3.2.1, the integrals in (11) and (12) can be computed analytically. More generally, standard numerical techniques such as Gauss-Hermite quadrature may provide reasonably fast and accurate solutions for prior families that do not result in closed-form integrals since the integrals are all one-dimensional. As a result, $\hat{\boldsymbol{\theta}}$ can often be computed efficiently using off-the-shelf optimization algorithms.

In summary, solving the cEBNM problem consists in finding a mapping from known quantities $(\hat{\boldsymbol{\beta}}, \boldsymbol{s}, \mathbf{D})$ to a tuple $(\hat{\boldsymbol{\theta}}, q)$, where each $(\boldsymbol{d}_i, \hat{\boldsymbol{\theta}})$ maps to an element $g(\boldsymbol{d}_i, \boldsymbol{\theta}) \in \mathcal{G}$, and $q$ is the posterior distribution of the unobserved $\boldsymbol{\beta}$ given $(\hat{\boldsymbol{\beta}}, \boldsymbol{s}, \mathbf{D})$. We denote this mapping as

$$\text{cEBNM}(\hat{\boldsymbol{\beta}}, \boldsymbol{s}, \mathbf{D}) = (\hat{\boldsymbol{\theta}}, q). \tag{13}$$

In practice, the full posterior $q$ is not needed; the first and second posterior moments are sufficient (see Sec. 3.3.2). Any prior family is admissible under the cEBMF framework so long as 13 is computable.

### 3.3.2 ALGORITHM

Given a method for solving the cEBNM problem (Sec. 3.3.1), the cEBMF model can be fitted using a simple coordinate ascent algorithm. In brief, the cEBMF algorithm maximizes an objective function—the evidence lower bound (ELBO) (Blei et al., 2017) under a variational approximation with conditional independence assumptions on $\mathbf{L}$ and $\mathbf{F}$—by iterating over the following updates for each factor $k = 1, \ldots, K$ until some stopping criterion is met:

1. Remove the effect of the $k$-th factor from the $n \times p$ matrix $\bar{\mathbf{R}}$ of expected residuals:

$$\bar{\mathbf{R}}^k = \bar{\mathbf{R}} - \bar{\boldsymbol{\ell}}_k \bar{\boldsymbol{f}}_k^T. \tag{14}$$

2. For each $i = 1, \ldots, n$, compute the least-squares estimates of $\ell_{ik}$, denoted $\hat{\ell}_{ik}$, and the standard deviations $s_{ik}^\ell$ of these estimates,

$$\hat{\ell}_{ik} = \sum_{j=1}^p \tau_{ij} r_{ij}^k \bar{f}_{jk} / (s_{ik}^\ell)^2 \tag{15}$$

$$s_{ik}^\ell = (\sum_{j=1}^p \tau_{ij} \bar{f}_{jk}^2)^{-1/2}, \tag{16}$$

where $\bar{f}_{jk}$ and $\bar{f}_{jk}^2$ denote, respectively, the first and second posterior moment of $f_{jk}$.

3. Update the parameter $\boldsymbol{\theta}$ that maps each individual vector of covariates $\boldsymbol{x}_i$ to a prior $g_k^{(\ell)}(\boldsymbol{x}_i, \boldsymbol{\theta}) \in \mathcal{G}_{\ell,k}$ by solving (10), in which we make the following substitutions:

$$\hat{\beta}_i \leftarrow \hat{\ell}_{ik}, \quad s_i \leftarrow s_{ik}^\ell, \quad i = 1, \ldots, n, \quad \mathbf{D} \leftarrow \mathbf{X}, \quad \mathcal{G} \leftarrow \mathcal{G}_{\ell,k}. \tag{17}$$

4. Making the same substitutions in (12), update the posterior means $\bar{\boldsymbol{\ell}}_k$ and posterior second moments $\bar{\boldsymbol{\ell}}_k^2$.

5. Perform updates similar to those in Steps 2–4 to update $\bar{\boldsymbol{f}}_k$, $\bar{\boldsymbol{f}}_k^2$ and $g_k^{(f)} \in \mathcal{G}_{f,k}$.

6. Update the matrix of expected residuals by adding back the effect of the $k$-th factor:

$$\bar{\mathbf{R}} = \bar{\mathbf{R}}^k + \bar{\boldsymbol{\ell}}_k \bar{\boldsymbol{f}}_k^T. \tag{18}$$

Also, given initial estimates of $\bar{\mathbf{L}}, \bar{\mathbf{F}}$, the expected residuals are initialized as $\bar{\mathbf{R}} = \mathbf{Z} - \bar{\mathbf{L}}\bar{\mathbf{F}}^T$. To simplify the presentation, we have omitted some details here, such as how to update the residual variances $\tau_{ij}^{-1}$. See the Appendix for a fuller description and derivation.

**Computational complexity.** Since cEBMF is a framework, not a specific method, we cannot give the exact computational complexity of this algorithm, but we can provide some guidelines. Steps 1, 2 and 6 involve preparing the inputs to the cEBNM solver (13). Since these steps do not depend on the prior families $\mathcal{G}_{\ell,k}, \mathcal{G}_{f,k}$, we can give their computational complexity: when $\mathbf{Z}$ is a "dense" (non-sparse) matrix, the time complexity is $O(np)$; when $\mathbf{Z}$ is sparse, the complexity is $O(S)$ (with careful implementation), where $S$ is the number of nonzero entries in $\mathbf{Z}$. Steps 3–5 will depend on the particular cEBNM solver and the type of side information. However, when the priors on $\mathbf{L}, \mathbf{F}$ are simple and involve low-dimensional covariates, Steps 1, 2 and 6 will dominate, and so cEBMF should be able to handle large data sets in this setting.

## 4 EXPERIMENTS

### 4.1 SIMULATIONS

First we compared cEBMF with other matrix factorization methods in simulated data sets. We compared cEBMF to two methods that do not use the side information—EBMF (flashier R package; Wang and Stephens 2021) and the penalized matrix decomposition, "PMD" (PMA R package; Witten et al. 2009)—and three methods that use side information: MFAI (mfair R package; Wang et al.

2024), collective matrix factorization, "CMF" (cfmrec R package; Singh and Gordon 2008), and Spatial PCA (SpatialPCA R package; Shang and Zhou 2022). Spatial PCA accepts only a specific type of side information, the 2-d coordinates of the data points. For all methods, we set the rank of the matrix factorization to the $K$ that was used to simulate the data. The penalty parameters in PMD were tuned via cross-validation as recommended by the authors. For EBMF and cEBMF, we assumed homogeneous noise, $\tau_{ij} = \tau$, and we chose prior families appropriate for each simulation setting (except for the "mis-specified prior" simulation; see below). These prior families were all slightly more flexible versions of the "spike and slab" priors in Sec. 3.2.1. The parameterized priors in cEBMF were either single-layer neural networks with softmax activation function (multinomial regression) or a multi-layer perceptron. The parameters of the neural network were learned using the Keras R interface for TensorFlow (Abadi et al., 2016). (More details are given in the Appendix.)

We simulated data sets under the following settings.

**Sparsity-driven covariate.** This simulation was intended to illustrate the behaviour of cEBMF when provided with simple row- and column-covariates that inform only the sparsity of $\mathbf{L}$ and $\mathbf{F}$ (and not the magnitudes of their elements). The side information was stored in $1{,}000 \times 10$ and $200 \times 10$ matrices $\mathbf{X}$ and $\mathbf{Y}$, and the $1{,}000 \times 200$ matrix $\mathbf{Z}$ was simulated using a simple cEBMF model with $K = 2$, $\tau_{ij} = 0.25$, and with spike-and-slab priors chosen to ensure that 90% of the elements of the "denoised" matrix, $\mathbf{L}\mathbf{F}^T$, were zero. For EBMF, we assigned scale mixtures of normals with a fixed grid of scales (Stephens, 2017) as the prior family to all rows and columns of $\mathbf{L}$ and $\mathbf{F}$. For cEBMF, we used a prior family of the same form, except that the weights in the mixture were determined by covariates, $g(\boldsymbol{d}_i, \theta) = \pi_0(\boldsymbol{d}_i, \boldsymbol{\theta})\delta_0 + \sum_{m=1}^{M} \pi_m(\boldsymbol{d}_i, \boldsymbol{\theta})\mathcal{N}(0, \sigma_m^2)$.

**Uninformative covariate.** To verify that cEBMF was robust to situations in which the side information was not helpful, we considered an "uninformative covariate" setting in which the covariates were just noise. The data sets were simulated in the same way as the sparsity-driven covariate simulation, except that $\tau_{ij} = 1$ and the true factors were simulated as $\ell_{ik} \sim (1 - \pi_\ell)\delta_0 + \pi_\ell \mathcal{N}(0, 1)$, $f_{jk} \sim (1 - \pi_f)\delta_0 + \pi_f \mathcal{N}(0, 1)$, with the weights $\pi_\ell, \pi_f$ chosen to achieve a target sparsity of 90% zeros in the "denoised" matrix. EBMF and cEBMF were run in the same way as in the sparsity-driven covariate simulations.

**Tiled-clustering model.** In this setting, we simulated matrix factorizations in which $\mathbf{L}$ (but not $\mathbf{F}$) depended on the 2-d locations of the data points. Specifically, we generated a periodic tiling of $[0, 1] \times [0, 1]$, randomly labeling each tile 1, 2 or 3. (One of these simulations is shown in Fig. 1.) For each data point $i$, we set $\ell_{ik} = 1$ if $i$ was in the tile with label $k$, otherwise $\ell_{ik} = 0$. The $\mathbf{F}$ matrix, by contrast, was simulated from a simple scale mixture of normals, $f_{jk} \sim \pi_0\delta_0 + \sum_{m=1}^{M} N(0, \sigma_m^2)$. We simulated homoskedastic noise with $\tau_{ij} = 0.1$. Notice that $\mathbf{L}$ in this simulation was always non-negative. Therefore, we chose the prior families in EBMF to produce *semi-non-negative matrix factorizations* (Ding et al., 2010); specifically, point-exponential priors for $\mathbf{L}$—flexible priors similar to the spike-and-slab prior (Sec. 3.2.1) but with support for non-negative numbers only (Willwerscheid et al., 2024)—and scale mixtures of normal priors, the same as above, for $\mathbf{F}$. Analogously, the prior families for columns of $\mathbf{L}$ in cEBMF were scale mixtures of exponentials modulated by the row covariate, $g(\boldsymbol{d}_i, \boldsymbol{\theta}) = \pi_0(\boldsymbol{d}_i, \boldsymbol{\theta})\delta_0 + \sum_{m=1}^{M} \pi_m(\boldsymbol{d}_i, \boldsymbol{\theta}) \exp(\lambda_m)$, where $\exp(\lambda)$ is the exponential distribution with scale parameter $\lambda$, and $\lambda_{m-1} < \lambda_m$, $m = 2, \ldots, M$. The prior families for $\mathbf{F}$ were the same as in EBMF.

**Mis-specified prior—"shifted tiled-clustering" model.** To further asses robustness of the methods, we performed an additional set of simulations to assess how well cEBMF (and EBMF) would deal with situations in which the prior was mis-specified; that is, when no prior within the chosen prior family could recover the true data generation process. These simulations were similar to the tiled-clustering model except that we generated the $i$-th row of $\mathbf{L}$ as follows: $(1, 2, 3)$ if data point was $i$ in the tile with label 1; $(3, 1, 2)$ if data point was in the tile with label 2, and $(2, 3, 1)$ if data point was in the tile with label 3. We then ran the methods in the same way as in the tiled-clustering simulations.

We simulated 100 data sets in each of these settings. To evaluate the matrix factorizations, we computed the root mean squared error (RMSE) between the true factorization $\mathbf{L}\mathbf{F}^T$ and the estimated factorization $\hat{\mathbf{L}}\hat{\mathbf{F}}^T$.

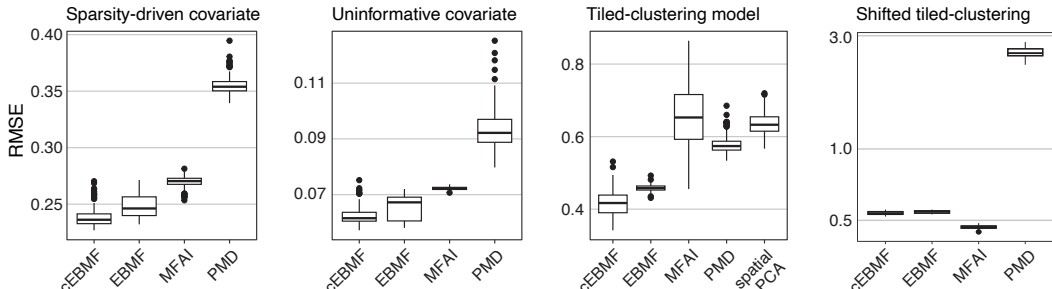

Figure 2: Performance of different matrix factorizations in the simulated data sets. The boxplots summarize the error (RMSE) in the matrix factorizations across 100 simulations (lower RMSEs are better). See Figures 5–8 in the Appendix for additional simulation results. Note that Fig. 1 shows results from one of the tiled-clustering simulations in greater detail.

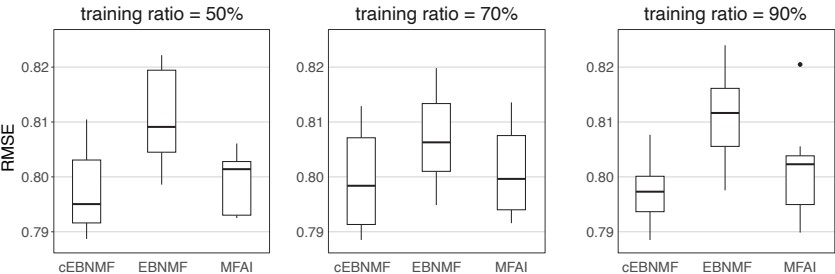

Figure 3: Prediction performance of different matrix factorization methods in the MovieLens 100K data (Harper and Konstan, 2015), with varying degrees of the movie ratings available for training. Training ratio = R% means that R% of the movie ratings were used in training, and the remaining $(100 - R)\%$ movie ratings were used to evaluate accuracy of the matrix factorization (measured using RMSE). The results at each training ratio are from 30 randomly generated training/test sets.

The results of these simulations are summarized in Fig. 2. (Additional results, including comparisons with other methods, including CMF, are given in the Appendix.) In the first three simulation settings, cEBMF produced matrix factorizations that were more accurate than, or at least as accurate as, the other methods. Reassuringly, cEBMF was no worse than EBMF in the uninformative covariate and shifted tiled-clustering settings. (Although we caution that cEBMF may "overfit" the prior when the covariates are mostly noise; see Sec. 5 for a brief discussion of this point.) cEBMF achieved the greatest gains over EBMF in the tiled-clustering setting where the covariates were also the most informative. The poor performance of Spatial PCA in this setting illustrates a key point: although Spatial PCA leveraged the side information to improve accuracy, it is not ideally suited to this setting due to the "PCA-like" (orthogonality) constraints imposed on the matrix factorization (Shang and Zhou, 2022). A more flexible framework with priors that can be tailored to a particular setting will generally have an advantage. (To be fair, Spatial PCA is perhaps better suited to spatial transcriptomics data sets than these simulated data sets.) The siulations generally did not conform well to MFAI's modeling assumptions, and indeed MFAI performed much worse than both cEBMF and EBMF. The one exception was the shifted tiled-clustering setting where the modeling assumptions of all methods were wrong—but MFAI's appeared to be the least wrong.

## 4.2 COLLABORATIVE FILTERING

To provide a quantitative comparison of the matrix factorization methods on real data, we ran EBMF, cEBMF and MFAI on the on the MovieLens 100K data (Harper and Konstan, 2015). In the Movie-Lens data, $\mathbf{Z}$ is a $1,682 \times 943$ matrix, with rows corresponding to movies and columns corresponding to users. Since most movie trainings are missing, this data set also serves illustrate the ability of cEBMF (as well as EBMF and MFAI) to seemlessly handle missing data. The side information $\mathbf{X}$ used by cEBMF and MFAI was a $1,682 \times 19$ binary matrix with columns corresponding to movie genres. We randomly hid different proportions of movie ratings from the training set, then we as-

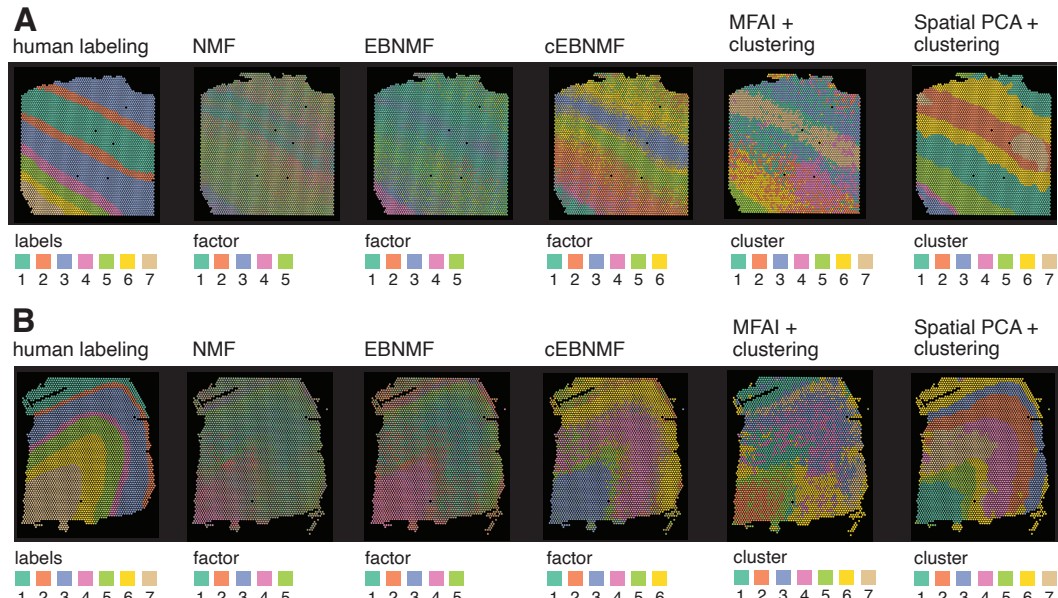

Figure 4: Results on slides 4 (A) and 10 (B) of the DLPFC spatial transcriptomics data (Pardo et al., 2022). For the NMF, EBMF and cEBMF results, each pixel $i$ is shown as a pie chart using the relative values of $i$-th row of $\mathbf{L}$ (after performing an "LDA-style" post-processing of $\mathbf{L}$, $\mathbf{F}$; Townes and Engelhardt 2023). Higher-resolution versions of these images are included in the supplementary ZIP file. CMF results and additional results on all 12 slices are given in the Appendix.

sessed how well the estimated matrix factorization $\hat{\mathbf{L}}\hat{\mathbf{F}}^T$ recovered the hidden ratings. (Note that although several larger MovieLens data sets exist, and we successfully applied EBMF and cEBMF to larger MovieLens data sets, we could not get MFAI to run on the larger data sets.)

We ran EBMF and cEBMF so as to produce non-negative matrix factorizations, which is common in collaborative filtering (e.g., Singh and Gordon 2008). (Therefore, in the results we labeled these methods as "EBNMF" and "cEBNMF".) To enforce non-negativity in $\mathbf{L}$ and $\mathbf{F}$, we used the same scale mixtures of exponentials that were used for $\mathbf{L}$ in the tiled-clustering simulations. The mixture weights in the cEBNMF prior were implemented as a multi-layer perceptron. (Note that MFAI cannot produce non-negative matrix factorizations.) Rather than fix the number of factors, $K$, like we did in the simulations, we let all methods choose $K$; the idea is that the methods should adapt $K$ automatically to the complexity of the data. We upper bounded $K$ at 7 for EBNMF and cEBNMF, and 12 for MFAI.

Figure 3 summarizes the results of running the different matrix factorization methods on the Movie-Lens 100K data sets with different fractions of the movie ratings available for training. Both MFAI and cEBNMF were able to use the side information (the movie genres) to provide improvements over EBNMF, and cEBNMF tended to produce the largest improvements in accuracy.

## 4.3 SPATIAL TRANSCRIPTOMICS

Although cEBMF was not specifically designed with spatial data in mind, here we show that cEBMF also yields compelling results from spatial transcriptomics data (Marx, 2021). We illustrate this using a data set (Pardo et al., 2022) that has been annotated by domain experts (Maynard et al., 2021) and has been used in several papers to benchmark methods for spatial transcriptomics (e.g., Shang and Zhou 2022; Zhao et al. 2021; Zhu et al. 2023). The data were collected from 12 slices of the human dorsolateral prefrontal cortex (DLPFC) tissue. After data preprocessing, each slice contained about 4,000 pixels and expression measured in about 5,000 genes ($n \approx 4000$, $p \approx 5000$).

Our aim was to generate a "parts-based" representation of the data, with the hopes that the "parts" would resolve to biologically meaningful units (e.g., cell types, tissue regions) (Townes and Engel-

hardt, 2023). With this aim in mind, we ran cEBMF so as to produce non-negative matrix factorizations ("cEBNMF") using the same priors as in the MovieLens data set. We compared cEBNMF to CMF, MFAI and Spatial PCA, and two other non-negative matrix factorizations that did not exploit the spatial data—NMF (implemented in the R package NNLM; Lin and Boutros 2020) and EBMF with point-exponential priors ("EBNMF"). Although Spatial PCA cannot be directly compared to NMF, the Spatial PCA software also clusters the data points after projection onto the principal components (PCs), and this clustering can be compared to the non-negative matrix factorizations. Following Shang and Zhou (2022), we computed the top 20 PCs, then we ran the walk-trap clustering algorithm (Pons and Latapy, 2005) on the PCs. Since MFAI also does not produce a non-negative matrix factorization, we clustered the MFAI output in a similar way to Spatial PCA. Additional details are given in the Appendix.

Figure 4 shows the results on two of the slices, with results on all 12 slices given in the Appendix. Qualitatively, some of the factors from NMF and EBMF seem to correspond to the expert-labeled regions, but several other factors appear to be capturing other substructures that have no obvious spatial quality. Comparatively, the cEBNMF results in slices 4 and 9 capture the expert labeling much more closely, with most factors showing a clear spatial quality. The clusters obtained from the Spatial PCA and MFAI factorizations also capture spatial structure and expert labeling well, with some exceptions, e.g., Spatial PCA cluster 7 in slices 4 and 9. (The Spatial PCA software performed an additional post-processing step on the clusters which is why these clusters look less noisy.) The CMF results were very poor, reflecting the inappropriateness of the CMF assumptions in this setting (Fig. 9 in the Appendix). Note that the NMF methods can capture continuous variation in expression within and across cell types or regions—as well as the expectation that some pixels might be combinations of cell types—whereas the clustering cannot.

## 5 LIMITATIONS

**Overfitting.** Empirical Bayes methods are known to be prone to overfitting; this is of particular concern for more complex priors with many free parameters. Following Tansey et al. (2020), we found that adding a simple $L_2$-penalty term on the hyperparameters helped, but this is an area where our methods can be improved.

**Priors for more complex covariates.** The demonstrations of cEBMF in this paper all incorporated relatively simple types of side information, so it remains to be seen how well cEBMF can work with more complex covariates. In the spatial transcriptomics application, we tried using the histological images included in the DLPFC experiment as side information, but it seemed that the images provided in the spatialLIBD format (Pardo et al., 2022) were of too poor quality to be useful for inferring cell types or tissue regions.

## 6 CONCLUSIONS

We have introduced cEBMF, a general framework for matrix factorization in which (i) side information is incorporated via flexible priors, and (ii) the priors are learned from the data using empirical Bayes ideas. We have put considerable effort into optimizing the software implementation, in part by leveraging previous work in this area (Wang and Stephens, 2021; Willwerscheid et al., 2024; Willwersheid, 2021), so that we expect these methods to scale well to even larger data sets with, say, hundreds of thousands of rows and/or columns.

Our experiments underscored the importance of having a matrix factorization framework in which the model assumptions are appropriate for the target data set or sufficiently flexible that they can adapt to a variety of settings. The priors in cEBMF can be virtually any probabilistic model that can be optimized via (10–11). Our framework may suggest new ways to integrate other more complex types of side information such as images or graphs. Further, we envision that cEBMF could be used in conjunction with pre-trained models (e.g., convolutional neural networks for cell classification; Zhang et al. 2017) to reduce training time and improve accuracy in complex data sets.

**Note:** R code implementing the methods as well as the experiments is included in a supplementary ZIP file accompanying the manuscript.

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

## A  DETAILED METHODS AND DERIVATIONS

For convenience, we give the full cEBMF model here:

$$\mathbf{Z} = \mathbf{L}\mathbf{F}^T + \mathbf{E} \tag{19}$$

$$= \sum_{k=1}^{K} \boldsymbol{\ell}_k \boldsymbol{f}_k^T + \mathbf{E} \tag{20}$$

$$\ell_{ik} \sim g_k^{(\ell)}(\,\cdot\,;\boldsymbol{x}_i), g_k^{(\ell)}(\,\cdot\,;\boldsymbol{x}_i) \in \mathcal{G}_{\ell,k} \tag{21}$$

$$f_{jk} \sim g_k^{(f)}(\,\cdot\,;\boldsymbol{y}_j), g_k^{(f)}(\,\cdot\,;\boldsymbol{y}_j) \in \mathcal{G}_{f,k} \tag{22}$$

$$e_{ij} \sim N(0, \tau_{ij}^{-1}), \tag{23}$$

where the matrices $\mathbf{L}$ ($n \times K$) and $\mathbf{F}$ ($p \times K$) are both unknown and $\boldsymbol{f}_k \in \mathbb{R}^p$, $\boldsymbol{\ell}_k \in \mathbb{R}^n$ for $k = 1, \ldots, K$. $\mathbf{E}$ is an $n \times p$ matrix of errors where the precision parameters $\boldsymbol{\tau}$ are unknown. $\boldsymbol{x}_i$ denotes the information available for the $i$-th row of $\mathbf{Z}$ and $\boldsymbol{y}_j$ denotes the information available for the $j$-th column of $\mathbf{Z}$. We assume, for simplicity, that it is possible to store this information in matrix form, so that $\mathbf{X}$ is a $n \times p_{\mathbf{X}}$ matrix that contains information on the rows of $\mathbf{Z}$ and $\mathbf{Y}$ is a $p \times p_{\mathbf{Y}}$ matrix that contains information on the columns of $\mathbf{Z}$. $g_k^{(\ell)}(.;\boldsymbol{x}_i)$ is a probability distribution within the family $\mathcal{G}_{\ell,k}$ parameterized by $\boldsymbol{x}_i$, with $g_k^{(\ell)}(l;\boldsymbol{x}_i)$ denoting the density of $g_k^{(\ell)}(\,\cdot\,;\boldsymbol{x}_i)$ at point $l$. Our derivation essentially follows the seminal work of Wang and Stephens (2021). Our main contribution lies in incorporating covariates into the model (19)–(23) and in showing that a fitting procedure similar to the one proposed by Wang and Stephens (2021) can be used to fit a model that is substantially more complex than EBMF.

### A.1  LEMMA ON THE CEBNM MODEL

Before formally demonstrating that we can fit the cEBMF model using the cEBNM model, we introduce a simple lemma on the cEBNM model. We first recall the cEBNM model:

$$\hat{\beta}_i = \beta_i + e_i \tag{24}$$

$$\beta_i \sim g(\cdot; \boldsymbol{x}_i) \tag{25}$$

$$e_i \sim \mathcal{N}(0, s_i^2). \tag{26}$$

To lighten the derivation below, we abuse notation slightly and denote $g(\cdot; \boldsymbol{x}_i, \boldsymbol{\theta})$ by $g(\cdot; \boldsymbol{x}_i)$ and $g(\cdot, \hat{\boldsymbol{\theta}})$ by $\hat{g}(\cdot)$. Solving the cEBNM problem involves

1. Estimating $g$ by maximum likelihood,

$$\hat{g} = \arg\max_g \mathcal{L}(g), \tag{27}$$

where $\mathcal{L}(g) = \int \prod_i p(\hat{\beta}_i | \beta_i, s_i) g(\beta_i; \boldsymbol{x}_i) d\beta_i$, with $g(\beta_i; \boldsymbol{x}_i)$ corresponding to the density $g(\cdot; \boldsymbol{x}_i)$ computed at the point $\beta_i$.

2. Computing the posterior distributions

$$p(\beta_i | \hat{\beta}_i, \hat{g}, \boldsymbol{x}_i) \propto \prod_i p(\hat{\beta}_i | \beta_i, s_i) g(\beta_i; \boldsymbol{x}_i). \tag{28}$$

In brief, solving the cEBNM problem consists in finding a mapping from known quantities $(\hat{\boldsymbol{\beta}}, \boldsymbol{s}, \boldsymbol{X})$ to estimated quantities $(\hat{\boldsymbol{\theta}}, q)$, where $\hat{\boldsymbol{\theta}}$ maps a set of covariates $\boldsymbol{x}_i$ to a prior $g(\cdot; \boldsymbol{x}_i)$ and $q$ is the posterior distribution of the unobserved $\boldsymbol{\beta}$ given $(\hat{\boldsymbol{\beta}}, \boldsymbol{s}, \boldsymbol{X})$. We denote this mapping as cEBNM,

$$\text{cEBNM}(\hat{\boldsymbol{\beta}}, \boldsymbol{s}, \boldsymbol{X}) = (\hat{g}, q). \tag{29}$$

**Lemma A.1.** Solving the cEBNM problem also maximizes $\mathcal{F}^{\mathrm{cNM}}(q_{\boldsymbol{\beta}}, g, \mathbf{X})$ over $(q_{\boldsymbol{\beta}}, g)$, where

$$\mathcal{F}^{\mathrm{cNM}}(q_{\boldsymbol{\beta}}, g, \mathbf{X}) = \mathbb{E}_{q_{\boldsymbol{\beta}}}\left[\frac{1}{2}\sum_i (A_i \beta_i^2 - 2B_i \beta_i)\right] + \mathbb{E}_{q_{\boldsymbol{\beta}}}\left[\log \frac{p(\boldsymbol{\beta}|g, \mathbf{X})}{q_{\boldsymbol{\beta}}(\boldsymbol{\beta})}\right], \tag{30}$$

with $A_i = 1/s_i^2$, $B_i = \hat{\beta}_i/s_i^2$ and $q_{\boldsymbol{\beta}}(\boldsymbol{\beta}) = \prod_i q_{\beta_i}(\beta_i)$.

*Proof.* The cEBNM log-likelihood can be written as

$$\mathcal{L}(g, \hat{\boldsymbol{\beta}}, \mathbf{X}) = \log p(\hat{\boldsymbol{\beta}}|g, \mathbf{X}) \tag{31}$$

$$= \log\left[p(\hat{\boldsymbol{\beta}}, \boldsymbol{\beta}|g, \mathbf{X})/p(\boldsymbol{\beta}|\hat{\boldsymbol{\beta}}, g, \mathbf{X})\right] \tag{32}$$

$$= \int q_{\boldsymbol{\beta}}(\boldsymbol{\beta}) \log \frac{p(\hat{\boldsymbol{\beta}}, \boldsymbol{\beta}|g, \mathbf{X})}{p(\boldsymbol{\beta}|\hat{\boldsymbol{\beta}}, g, \mathbf{X})} d\boldsymbol{\beta} \tag{33}$$

$$= \int q_{\boldsymbol{\beta}}(\boldsymbol{\beta}) \log \frac{p(\hat{\boldsymbol{\beta}}, \boldsymbol{\beta}|g, \mathbf{X})}{q_{\boldsymbol{\beta}}(\boldsymbol{\beta})} d\boldsymbol{\beta} + \int q_{\boldsymbol{\beta}}(\boldsymbol{\beta}) \log \frac{q_{\boldsymbol{\beta}}(\boldsymbol{\beta})}{p(\boldsymbol{\beta}|\hat{\boldsymbol{\beta}}, g, \mathbf{X})} d\boldsymbol{\beta} \tag{34}$$

$$= \mathcal{F}^{\mathrm{cNM}}(q_{\boldsymbol{\beta}}, g, \mathbf{X}) + D_{\mathrm{KL}}\left(q_{\boldsymbol{\beta}}||p(\boldsymbol{\beta}|\hat{\boldsymbol{\beta}}, g, \mathbf{X})\right), \tag{35}$$

where

$$\mathcal{F}^{\mathrm{cNM}}(q_{\boldsymbol{\beta}}, g, \mathbf{X}) = \int q_{\boldsymbol{\beta}}(\boldsymbol{\beta}) \log \frac{p(\hat{\boldsymbol{\beta}}, \boldsymbol{\beta}|g, \mathbf{X})}{q_{\boldsymbol{\beta}}(\boldsymbol{\beta})} d\boldsymbol{\beta} \tag{36}$$

and $D_{\mathrm{KL}}(q||p)$ denotes the Kullback-Leibler (KL) divergence from distribution $p$ to distribution $q$.

By rearranging (31) and (35) we can write $\mathcal{F}^{\mathrm{cNM}}$ as the difference between the log-likelihood and the KL divergence,

$$\mathcal{F}^{\mathrm{cNM}}(q_{\boldsymbol{\beta}}, g, \mathbf{X}) = \mathcal{L}(g, \hat{\boldsymbol{\beta}}, \mathbf{X}) - D_{\mathrm{KL}}\left(q_{\boldsymbol{\beta}}(\boldsymbol{\beta})||p(\boldsymbol{\beta}|\hat{\boldsymbol{\beta}}, g, \mathbf{X})\right). \tag{37}$$

Since $D_{\mathrm{KL}}(q||p) \geq 0$ with equality when $p = q$, $\mathcal{F}^{\mathrm{cNM}}(q_{\boldsymbol{\beta}}, g, \mathbf{X})$ is maximized over $q_{\boldsymbol{\beta}}$ by setting $q_{\boldsymbol{\beta}}(\boldsymbol{\beta}) = p(\boldsymbol{\beta}|\hat{\boldsymbol{\beta}}, g, \mathbf{X})$; further,

$$\max_{q_{\boldsymbol{\beta}}} \mathcal{F}^{\mathrm{cNM}}(q_{\boldsymbol{\beta}}, g, \mathbf{X}) = \mathcal{L}(g, \hat{\boldsymbol{\beta}}, \mathbf{X}). \tag{38}$$

To complete the proof, notice that

$$\log p(\hat{\boldsymbol{\beta}}, \boldsymbol{\beta}|g, \mathbf{X}) = \frac{1}{2}\sum_i s_i^{-2}(\hat{\beta}_i - \beta_i)^2 + \log(p(\beta_i|g, \boldsymbol{x}_i)) + \textit{const.}, \tag{39}$$

so that

$$\mathcal{F}^{\mathrm{cNM}}(q_{\boldsymbol{\beta}}, g, \mathbf{X}) = \mathbb{E}_{q_{\boldsymbol{\beta}}}\left[\frac{1}{2}\sum_i s_i^{-2}(\hat{\beta}_i - \beta_i)^2\right] + \sum_i \mathbb{E}_{q_{\boldsymbol{\beta}}}\left[\log \frac{p(\beta_i|g, \boldsymbol{x}_i)}{q_{\beta_i}(\beta_i)}\right] + \textit{const.} \tag{40}$$

$\square$

## A.2 VARIATIONAL APPROXIMATION AND ELBO

Letting $q_{\boldsymbol{L}}$ and $q_{\boldsymbol{F}}$ be the variational distribution on the $K$ factors, we assume the following factorizations:

$$q_{\boldsymbol{L}} = \prod_k \prod_i q_{\ell_{ik}}(\ell_{ik}) \tag{41}$$

$$q_{\boldsymbol{F}} = \prod_k \prod_j q_{f_{jk}}(f_{jk}) \tag{42}$$

Then the objective function of the cEBMF model (19)–(23) is a function of $q_{\boldsymbol{L}} = (q_{\boldsymbol{\ell}_1}, \ldots, q_{\boldsymbol{\ell}_K})$, $q_{\boldsymbol{F}} = (q_{\boldsymbol{f}_1}, \ldots, q_{\boldsymbol{f}_K})$, $g_{\boldsymbol{F}} = (g_1^{(f)}, \ldots, g_K^{(f)})$, $g_{\boldsymbol{L}} = (g_1^{(\ell)}, \ldots, g_K^{(\ell)})$ and $\boldsymbol{\tau}$:

$$
\text{ELBO}(q_{\boldsymbol{L}}, q_{\boldsymbol{F}}, g_{\boldsymbol{F}}, g_{\boldsymbol{L}}, \boldsymbol{\tau}) = \int \prod_k q_{\ell k}(\boldsymbol{\ell}_k) q_{\boldsymbol{f}_k}(\boldsymbol{f}_k) \log \left( \frac{p(\mathbf{Z}, \boldsymbol{L}, \boldsymbol{F} | g_k^{(\ell)}, g_k^{(f)}, \boldsymbol{\tau}, \mathbf{X}, \mathbf{Y})}{q_{\ell k}(\boldsymbol{\ell}_k) q_{\boldsymbol{f} k}(\boldsymbol{f}_k)} \right) d\boldsymbol{\ell}_k d\boldsymbol{f}_k
$$
(43)

$$
= \mathbb{E}_{q_{\boldsymbol{L}}, q_{\boldsymbol{F}}}[\log(p(\mathbf{Z}|\boldsymbol{L}, \boldsymbol{F}, \boldsymbol{\tau})] + \sum_k \mathbb{E}_{q_{\ell_k}} \left[ \log \frac{p(\boldsymbol{\ell}_k | g_k^{(\ell)}, \mathbf{X})}{q_{\ell_k}(\boldsymbol{\ell}_k)} \right] + \sum_k \mathbb{E}_{q_{\boldsymbol{f}_k}} \left[ \log \frac{p(\boldsymbol{f}_k | g_k^{(f)}, \mathbf{Y})}{q_{\boldsymbol{f} k}(\boldsymbol{f}_k)} \right]
$$
(44)

### A.2.1 CONNECTING THE K -FACTOR CEBMF MODEL TO THE CEBNM PROBLEM

**Proposition A.2.** For the $K$-factor cEBMF model, the maximization problem over $q_{\ell_k}, g_{\ell_k}$ and $g_{\boldsymbol{f}_k}, q_{\boldsymbol{f}_k}$ of the ELBO can be solved using a cEBNM mapping. More precisely,

$$
\underset{q_{\ell_k}, g_{\ell_k}}{\text{argmax}}(\text{ELBO}) = \text{cEBNM}(\hat{\boldsymbol{\ell}}(\mathbf{R}^k, \bar{\boldsymbol{f}}_k, \bar{\boldsymbol{f}}^2{}_k, \boldsymbol{\tau}), \mathbf{s}_{\boldsymbol{\ell}}(\bar{\boldsymbol{f}}^2{}_k, \boldsymbol{\tau}), \mathbf{X}),
$$
(45)

and similarly

$$
\underset{q_{\boldsymbol{f}_k}, g_{\boldsymbol{f}_k}}{\text{argmax}}(\text{ELBO}) = \text{cEBNM}(\hat{\boldsymbol{f}}(\mathbf{R}^k, \bar{\boldsymbol{\ell}}_k, \bar{\ell}^2{}_k, \boldsymbol{\tau}), \mathbf{s}_{\boldsymbol{f}}(\bar{\ell}^2{}_k, \boldsymbol{\tau}), \mathbf{Y}),
$$
(46)

where $\hat{l}$ and $\hat{f}$ are defined as

$$
\hat{\boldsymbol{\ell}}(\mathbf{Z}, \boldsymbol{\nu}, \boldsymbol{\omega}, \boldsymbol{\tau})_i = \frac{\sum_j \tau_{ij} z_{ij} \nu_j}{\sum_j \tau_{ij} \omega_j} \qquad \text{and} \qquad \hat{\boldsymbol{f}}(\mathbf{Z}, \boldsymbol{\nu}, \boldsymbol{\omega}, \boldsymbol{\tau})_j = \frac{\sum_i \tau_{ij} z_{ij} \nu_i}{\sum_i \tau_{ij} \omega_i}
$$
(47)

and $\mathbf{s}_{\boldsymbol{\ell}}$ and $\mathbf{s}_{\boldsymbol{f}}$ are defined as

$$
\mathbf{s}_{\boldsymbol{\ell}}(\boldsymbol{\omega}, \boldsymbol{\tau})_i = \left( \sum_j \tau_{ij} \omega_j \right)^{-0.5} \qquad \text{and} \qquad \mathbf{s}_{\boldsymbol{f}}(\boldsymbol{\omega}, \boldsymbol{\tau})_j = \left( \sum_i \tau_{ij} \omega_i \right)^{-0.5}.
$$
(48)

$\bar{\boldsymbol{f}}_k, \bar{\boldsymbol{f}}^2{}_k$ correspond to the first and second moment under $q_{\boldsymbol{f}_k}$ (resp $\bar{\boldsymbol{\ell}}_k, \bar{\ell}^2{}_k$), and $\mathbf{R}^k$ is the partial residual matrix defined as

$$
r_{ij}^k = z_{ij} - \sum_{k' \neq k} \bar{l}_{ik'} \bar{f}_{jk'}.
$$
(49)

*Proof.* Using (44), we write the objective function of the full model relative to row factor $\boldsymbol{\ell}_k$ as

$$
\text{ELBO}(q_{\ell_k}, g_k^{(\ell)}) = \mathbb{E}_{q_{\ell k}} \left[ -\frac{1}{2} \sum_i (A_{ik} l_{ik}^2 - 2 B_{ik} l_{ik}) \right] + \mathbb{E}_{q_{\ell_k}} \left[ \log \frac{p(\boldsymbol{\ell}_k | g_k^{(\ell)}, \mathbf{X})}{q_{\ell_k}(\boldsymbol{\ell}_k)} \right] + const.,
$$
(50)

where *const.* is constant with respect to $q_{\ell_k}, g_{\ell_k}$ and

$$
A_{ik} = \sum_j \tau_{ij} \mathbb{E}_{q_{\boldsymbol{f}_k}} \left[ \boldsymbol{f}_{jk}^2 \right]
$$
(51)

$$
B_{ik} = \sum_j \tau_{ij} \left( R_{ij}^k \mathbb{E}_{q_{\boldsymbol{f}_k}} \left[ \boldsymbol{f}_{jk} \right] \right).
$$
(52)

Equation (45) thus follows from Lemma A.1, with (46) proved similarly. $\qquad \square$

### A.3 UPDATES FOR RESIDUAL PRECISION PARAMETERS

Denoting the full variational approximation as $q(\cdot)$ and focusing on the part of ELBO that depend on $\boldsymbol{\tau}$, we have

$$\text{ELBO}(\boldsymbol{\tau}) = \mathbb{E}_q \sum_{i,j} \left( 0.5 \log(\tau_{ij}) - 0.5\tau_{ij}(z_{ij} - \sum_k l_{ik}f_{jk})^2 \right) + const \tag{53}$$

$$= 0.5 \sum_{i,j} \left[ \log(\tau_{ij}) + \tau_{ij}\bar{r}_{ij}^2 \right] + const., \tag{54}$$

where we define $\bar{r}_{ij}^2$ as

$$\bar{r}_{ij}^2 = \mathbb{E}_q[(z_{ij} - \sum_k l_{ik}f_{jk})^2] \tag{55}$$

$$= \left( z_{ij} - \sum_k \mathbb{E}_q(l_{ik})\mathbb{E}_q(f_{jk}) \right)^2 - \sum_k \mathbb{E}_q(l_{ik})^2 \mathbb{E}_q(f_{jk})^2 + \sum_k \mathbb{E}_q(l_{ik}^2)\mathbb{E}_q(f_{jk}^2) \tag{56}$$

$$= (z_{ij} - \sum_k \bar{l}_{ik}\bar{f}_{jk})^2 - \sum_k (\bar{l}_{ik})^2(\bar{f}_{jk})^2 + \sum_k \bar{l^2}_{ik}\bar{f^2}_{jk}. \tag{57}$$

Given the first and second posterior moments of $\boldsymbol{L}$ and $\boldsymbol{F}$, $\boldsymbol{\tau}$ can thus be estimated as

$$\hat{\boldsymbol{\tau}} = \arg\max \sum_{i,j} \left[ \log(\tau_{ij}) + \tau_{ij}\bar{r}_{ij}^2 \right].$$

If the variance is assumed to be column-specific (i.e., $\tau_{ij} = \tau_j$ for $j = 1, \ldots, p$), this leads to

$$\hat{\tau}_j = \frac{n}{\sum_i \bar{r}_{ij}^2}.$$

### A.4 CHOICE OF K

A noteworthy aspect of empirical Bayes methods for matrix factorization, highlighted in Bishop (1999), is their inherent ability to automatically determine $K$ Bishop (1999); Stegle et al. (2012); Wang and Stephens (2021). This feature comes from the fact that the maximum-likelihood estimates for $g_k^{(\ell)}$ and $g_k^{(f)}$ can converge to a point-mass at zero (assuming that the families $\mathcal{G}_{\ell,k}$ and $\mathcal{G}_{f,k}$ include a point-mass at zero). Thus, if $K$ is initially set to a large value, certain row or column factors will converge to zero, which effectively "zeroes out" the corresponding component $k$. Alternatively, in a stepwise or "greedy" approach where factors are introduced sequentially, the process halts upon introducing a factor that optimizes to zero. These two algorithms for fitting cEBMF with unknown $K$ ("backfitting" with $K$ taken larger than necessary and "greedily" attempting to add additional factors until failure) are summarized in Algorithms 2 and 3.

## B ALGORITHMS

### B.1 SINGLE-FACTOR UPDATE FOR CEBMF

The cEBMF algorithm outlined in Sec. 3.3.2 is described more formally in Algorithm 1.

---

**Algorithm 1** Single-factor update for rank-$K$ cEBMF model

---

**Require** A $(n \times p)$ data matrix $\boldsymbol{Z}$
**Require** Two matrices of covariates $\boldsymbol{X}$ $(n \times p_{\mathbf{X}})$ and $\boldsymbol{Y}$ $(p \times p_{\mathbf{Y}})$
**Require** A function $\mathrm{cEBNM}(\hat{\boldsymbol{\beta}}, \boldsymbol{s}, \boldsymbol{x})$ that solves the cEBNM problem (29)
**Require** Initial values for first moments $\bar{\boldsymbol{L}} := (\bar{\boldsymbol{\ell}}_1, \ldots, \bar{\boldsymbol{\ell}}_K)$, and $\bar{\boldsymbol{F}} := (\bar{\boldsymbol{f}}_1, \ldots, \bar{\boldsymbol{f}}_K)$
**Require** Initial values for second moments $\bar{\boldsymbol{L}}^2 := (\bar{\boldsymbol{\ell}}_1^2, \ldots, \bar{\boldsymbol{\ell}}_K^2)$, and $\bar{\boldsymbol{F}}^2 := (\bar{\boldsymbol{f}}_1^2, \ldots, \bar{\boldsymbol{f}}_K^2)$
**Require** An index $k$ indicating which factor to compute updated values for

1: Compute matrix of expected squared residuals $\bar{\boldsymbol{R}}^2$ using (57)
2: $\tau_j \leftarrow \frac{n}{\sum_i \bar{r}_{ij}^2}$ (for column-specific variances; can be modified to make other assumptions)
3: Compute partial residual matrix $\mathbf{R}^k = \mathbf{Z} - \sum_{k' \neq k} \bar{\boldsymbol{\ell}}_{k'} \bar{\boldsymbol{f}}_{k'}^T$
4: Compute $\hat{\boldsymbol{\ell}}(\mathbf{R}^k, \bar{\boldsymbol{f}}_k, \bar{\boldsymbol{f}^2}_k, \boldsymbol{\tau})$ and its standard error $\mathbf{s}_{\boldsymbol{\ell}}(\bar{\boldsymbol{f}^2}_k, \boldsymbol{\tau})$
5: $(\bar{\boldsymbol{\ell}}_k, \bar{\boldsymbol{\ell}}_k^2) \leftarrow \mathrm{cEBNM}(\hat{\boldsymbol{\ell}}, \boldsymbol{s}_{\boldsymbol{\ell}}, \mathbf{X})$
6: Compute $\hat{\boldsymbol{f}}(\mathbf{R}^k, \bar{\boldsymbol{\ell}}, \bar{\boldsymbol{\ell}^2}, \boldsymbol{\tau})$ and its standard error $\mathbf{s}_{\boldsymbol{f}}(\bar{\boldsymbol{\ell}^2}, \boldsymbol{\tau})$
7: $(\bar{\boldsymbol{f}}_k, \bar{\boldsymbol{f}}_k^2) \leftarrow \mathrm{cEBNM}(\hat{\boldsymbol{f}}, \boldsymbol{s}_{\boldsymbol{f}}, \mathbf{Y})$
8: **return** updated values $(\bar{\boldsymbol{\ell}}_k, \bar{\boldsymbol{\ell}}_k^2, \bar{\boldsymbol{f}}_k, \bar{\boldsymbol{f}}_k^2, \boldsymbol{\tau})$

---

## B.2 GREEDY ALGORITHM

---

**Algorithm 2** Greedy Algorithm for cEBMF

---

**Require** A $(n \times p)$ data matrix $\mathbf{Z}$
**Require** Two matrices of covariates $\boldsymbol{X}$ $(n \times p_{\mathbf{X}})$ and $\boldsymbol{Y}$ $(p \times p_{\mathbf{Y}})$
**Require** A function, $\mathrm{init}(\mathbf{Z}) \to (\boldsymbol{\ell}, \boldsymbol{f})$ that provides initial estimates for row factors and column factors (e.g., SVD)
**Require** A function $\mathrm{cEBNM}(\hat{\boldsymbol{\beta}}, \boldsymbol{s}, \boldsymbol{x})$ that solves the cEBNM problem (29)
**Require** A function **single update** $(\mathbf{Z}, \bar{\boldsymbol{\ell}}_k, \bar{\boldsymbol{\ell}}_k^2, \bar{\boldsymbol{f}}_k, \bar{\boldsymbol{f}^2}_k, \boldsymbol{\tau}, \mathbf{X}, \mathbf{Y}) \to (\bar{\boldsymbol{\ell}}, \bar{\boldsymbol{\ell}}_k^2, \bar{\boldsymbol{f}}_k, \bar{\boldsymbol{f}}_k^2, \boldsymbol{\tau})$

1: Initialize $K \leftarrow 0$
2: **repeat**
3:     $K \leftarrow K + 1$
4:     Compute matrix of expected residuals $\mathbf{R} = \mathbf{Z} - \sum_{k=1}^{K-1} \bar{\boldsymbol{\ell}}_k \bar{\boldsymbol{f}}_k^T$
5:     Initialize first moments $(\bar{\boldsymbol{\ell}}_K, \bar{\boldsymbol{f}}_K) \leftarrow \mathit{init}(\mathbf{R})$
6:     Initialize second moments by squaring first moments: $\bar{\boldsymbol{\ell}^2}_K \leftarrow (\bar{\boldsymbol{\ell}}_K)^2$; $\bar{\boldsymbol{f}^2}_K \leftarrow (\bar{\boldsymbol{f}}_K)^2$
7:     **repeat**
8:         $(\bar{\boldsymbol{\ell}}_K, \bar{\boldsymbol{\ell}}_K^2, \bar{\boldsymbol{f}}_K, \bar{\boldsymbol{f}}_K^2, \boldsymbol{\tau}) \leftarrow$ **single update** $(\mathbf{Z}, \bar{\boldsymbol{\ell}}_K, \bar{\boldsymbol{\ell}^2}_K, \bar{\boldsymbol{f}}_K, \bar{\boldsymbol{f}^2}_K, \boldsymbol{\tau}, \mathbf{X}, \mathbf{Y})$
9:     **until** converged
10: **until** $\bar{\boldsymbol{f}}_K$ or $\bar{\boldsymbol{\ell}}_K$ is 0
11: **return** $(\bar{\boldsymbol{\ell}}, \bar{\boldsymbol{\ell}^2}, \bar{\boldsymbol{f}}, \bar{\boldsymbol{f}^2}, \boldsymbol{\tau})$

---

## B.3 BACKFITTING ALGORITHM

---

**Algorithm 3** Backfitting algorithm for cEBMF

---

**Require** A $(n \times p)$ data matrix $\mathbf{Z}$
**Require** Two matrices of covariates $\boldsymbol{X}$ $(n \times p_{\mathbf{X}})$ and $\boldsymbol{Y}$ $(p \times p_{\mathbf{Y}})$
**Require** A function, $init(\mathbf{Z}) \rightarrow (\mathbf{L}, \mathbf{F})$ that provides initial estimates for row factors and column factors (e.g., SVD)
**Require** A function cEBNM$(\hat{\boldsymbol{\beta}}, \boldsymbol{s}, \boldsymbol{x})$ that solves the cEBNM problem (29)
**Require** A function **single update** $(\mathbf{Z}, \bar{\ell}_k, \bar{\ell}_k^2, \bar{f}_k, \bar{f}^2_k, \boldsymbol{\tau}, \mathbf{X}, \mathbf{Y}) \rightarrow (\bar{\ell}, \bar{\ell}_k^2, \bar{f}_k, \bar{f}_k^2, \boldsymbol{\tau})$
1: Initialize first moments $(\bar{\ell}_1, \ldots, \bar{\ell}_K; \bar{f}_1, \ldots, \bar{f}_K) \leftarrow init(\mathbf{Z})$
2: Initialize second moments by squaring first moments: $\bar{\ell}_k^2 \leftarrow (\bar{\ell}_k)^2; \bar{f}_k^2 \leftarrow (\bar{f}_k)^2$ for $k = 1, \ldots, K$
3: **repeat**
4:     **for** $k = 1, \ldots, K$ **do**
5:         $(\bar{\ell}_k, \bar{\ell}_k^2, \bar{f}_k, \bar{f}_k^2, \boldsymbol{\tau}) \leftarrow$ **single update** $(\mathbf{Z}, \bar{\ell}_k, \bar{\ell^2}_k, \bar{f}_k, \bar{f^2}_k, \boldsymbol{\tau}, \mathbf{X}, \mathbf{Y})$
6:     **end for**
7: **until** converged
8: **return** $(\bar{L}, \bar{L}^2, \bar{F}, \bar{F}^2, \boldsymbol{\tau})$

---

# C   DETAILS OF THE EXPERIMENTS

## C.1   ADDITIONAL METHODS COMPARISON

As mentioned in section 4 we comparde cEBMF with regular EBMF Wang and Stephens (2021), Penalized Matrix Decomposition Witten et al. (2009) (PMD, implemented in R package PMA) and when applicable, we also use Spatial PCA (SpaPCA) Shang and Zhou (2022). We also conducted comparison with Sparse SVD Yang et al. (2014) (SSVD, implemented in R package ssvd) and with standard PCA (implemented via truncated SVD). Below we detail the these experiments. We display the results of these numerical experiments in section D.

## C.2   METHODS PARAMETRIZATION

As mentioned in section 4, for all methods, we set the rank of the matrix factorization to the $K$ that was used to simulate the data. The penalty parameters in PMD were tuned via cross-validation as recommended by the authors. cEBMF, EBMF, and SpaPCA are 'self-tuning' methodologies and don't require fixing many hyperparameters beforehand. Following the comments of Yang et al. (2014), SSVD is robust to the choice of tuning parameters, so we ran SSVD with its default values. All the experiments were run using the *R* implementation of Keras, which is a wrapper for TensorFlow Abadi et al. (2016) functions.

## C.3   SPARSITY-DRIVEN COVARIATE SIMULATIONS

**Simulation procedure** In this simulation setting, we generate rank 2 covariate moderated factor model ($K = 2$ in 19) as follows. First, we simulate two sets of covariates $\mathbf{X}, \mathbf{Y}$ with 10 columns each. We simulate the entry of $\mathbf{X}$ by sampling iid realizations of $\mathcal{N}(0, 1)$ and similarly for the entries of $\mathbf{Y}$. Each of the columns of $\mathbf{X}$ and $\mathbf{Y}$ affect the sparsity of the row factor (resp column factor) via a a logistic model regression model

$$\ell_{ik} \sim \pi_0(\boldsymbol{x}_i, \boldsymbol{\theta}_k)\delta_0 + (1 - \pi_0(\boldsymbol{x}_i, \boldsymbol{\theta}_k))N(0, 1) \tag{58}$$

$$\boldsymbol{f}_{jk} \sim \pi_0(\boldsymbol{y}_j, \boldsymbol{\omega}_k)\delta_0 + (1 - \pi_0(\boldsymbol{y}_j, \boldsymbol{\omega}_k))N(0, 1) \tag{59}$$

$$\log\left(\frac{\pi_0(\boldsymbol{x}_i, \boldsymbol{\theta}_k)}{1 - \pi_0(\boldsymbol{x}_i, \boldsymbol{\theta}_k)}\right) = \boldsymbol{\theta}_k^T \boldsymbol{x}_i \tag{60}$$

$$\log\left(\frac{\pi_0(\boldsymbol{y}_j, \boldsymbol{\omega}_k)}{1 - \pi_0(\boldsymbol{y}_j, \boldsymbol{\omega}_k)}\right) = \boldsymbol{\omega}_k^T \boldsymbol{y}_j \tag{61}$$

The noise level $\sigma_{ij}^2 = 1/\tau_{ij}$ of $\mathbf{E}$ varies from 0.1 to 2. We select $\boldsymbol{\theta}_k$ and $\boldsymbol{\omega}_k$ such that the sparsity of $\sum_{k=1}^K \ell_k \boldsymbol{f}_k^T$ is equal to 90%.

We run cEBMF using priors of the form $\pi_0(.,\boldsymbol{\theta})\delta_0 + \sum_{m=1}^{M} \pi_m(.,\boldsymbol{\theta})N(0,\sigma_m^2)$ for both row and column factors, where $\boldsymbol{\pi}(.,\boldsymbol{\theta}) = (\pi_0(.,\boldsymbol{\theta}),\ldots,\pi_M(.,\boldsymbol{\theta}))$ is a multinomial regression model. We use a mixture of Gaussian prior for EBMF Stephens (2017). We fit cEBMF and EBMF using a homogeneous noise parameter $\sigma_{ij}^2 = \sigma^2$. We perform 400 simulations under each simulation configuration. The results are presented in Figure 2.

**Description of the cEBMF prior model the row and column factors.** We model the functions $g_k^{(\ell)}(.;.), g_k^{(f)}(.;.)$ using a single layer model with softmax activation function using 50 epoch per loop in Algorithm 2 and a batch size of 500 for the row factors and a batch size of 400 for the column factors.

### C.4    TILED-CLUSTERING MODEL

**Simulation procedure**    In this simulation setting we generated a rank 3 covariate moderated factor model matrix for ($K = 3$) as follow. First, we generate a periodic tilling of $[0,1]^2$. We use a simple periodic tiling of $[0,1]^2$ that divides the regions in nine square of width $\frac{1}{3}$.At each iteration we label at random three tiles with label 1, three other tiles with label 2 and the remaining tile with label 3.

We then sample uniformly some 1000 2d coordinates on $(x_i, y_i) \in [0,1]^2$. Then for each set of coordinate we set $\ell_{ik} = \mathbb{1}_{(x_i,y_i)\in\{\text{tile, label (tile)}=k\}}$ if $(x_i, y_i)$ in a tile with label $k$. The factors are sample under a Gaussian mixture $\boldsymbol{f}_{jk} \sim \pi_0\delta_0 + \sum_{m=1}^{M} N(0,\sigma_m^2)$. The noise level $\sigma_{ij}^2 = 1/\tau_{ij}$ of $\mathbf{E}$ varied from 1 to 5. We model the row factor prior in cEBMF using a model of the form $\pi_0(.,\boldsymbol{\theta})\delta_0 + \sum_{m=1}^{M} \pi_m(.,\boldsymbol{\theta})\exp(\lambda_m)$ where $\boldsymbol{\pi}(.,\boldsymbol{\theta}) = (\pi_0(.,\boldsymbol{\theta}),\ldots,\pi_M(.,\boldsymbol{\theta}))$ is a multi-layer perceptron (MLP), and $\exp(\lambda)$ is the density of the exponential distribution with parameter $\lambda$ and $\lambda_m < \lambda_{m+1}$ for all $m$. We model the column factors using a mixture of centered Gaussian as a prior for the column factor (with no covariate).

**Description of the cEBMF prior model the row factors.**    We use a sequential model with a dense layer with 64 units and ReLU activation. We use two subsequent dense layers, each with 64 units, and ReLU activation followed using an L2 regularization coefficient of 0.001 to prevent overfitting by penalizing large weights. Each of these regularized layers is followed by a dropout layer with a dropout rate of 0.5. The final Layer is a dense layer with a softmax activation. These model are trained using 50 epochs per loop in Algorithm 2 and a batch size of 500

The column factor parameters were estimated using the mixsqp *R* package which implements a sequential quadratic programming method for fast maximum-likelihood estimation of mixture proportions Kim et al. (2020). We fit cEBMF and EBMF using a homogeneous noise parameter $\sigma_{ij}^2 = \sigma^2$.

### C.5    SPATIAL TRANSCRIPTOMICS EXPERIMENTS

The spatial transcriptomic data were processed in the spatial PCA manuscript Shang and Zhou (2022) following the protocol proposed by the authors, which is available at `https://lulushang.org/SpatialPCA_Tutorial/DLPFC.html`. After processing the data using the normalization steps proposed by Shang and Zhou (2022), we ran cEBMF on each spatial transcript, parametrized as a covariate-moderated empirical Bayes non-negative factorization (cEBNMF). We parametrized cEBNMF model as follows. We use priors of the form $\pi_0(\cdot,\boldsymbol{\theta})\delta_0 + \sum_{m=1}^{M} \pi_m(\cdot,\boldsymbol{\theta})\exp(\lambda_m)$ for the row factors and a mixture of exponentials for the column factors. The location of each spot was used as a covariate for the row factors, modeled with an MLP ( i.e., $\boldsymbol{\pi}(.,\boldsymbol{\theta}) = (\pi_0(.,\boldsymbol{\theta}),\ldots,\pi_M(.,\boldsymbol{\theta}))$ is a MLP which we describe below) and set $K$ to the number of manually annotated clusters. We compared cEBNMF to EBNMF (using a mixture of point mass at zero and exponential components) and standard NMF (using the NNLM package Lin and Boutros (2020)) each run using K=5 as it results in better fit then for other K.

**Description of the cEBMF prior model the row factors.**    We use a sequential model with a dense layer with 64 units and ReLU activation. We use two subsequent dense layers, each with 64 units, and ReLU activation using an L2 regularization coefficient of 0.001 to prevent overfitting by penalizing large weights. These regularized layers are followed by a dropout layer with a dropout rate of 0.5. The subsequent layers are four dense layers, each with 64 units, and ReLU activation

using an L2 regularization coefficient of 0.001 The final Layer is a dense layer with a softmax activation. These model are trained using 300 epochs per loop in Algorithm 2 and a batch size of 1500.

The column factor parameters were estimated using the mixsqp package for fast maximum-likelihood estimation of mixture proportionsKim et al. (2020). We fit cEBMF and EBMF using a row-wise homogeneous noise parameter $\sigma_{ij}^2 = \sigma_j^2$.

## C.6 COMPUTING AND SOFTWARE NOTES

All the experiments were run using a high-performance computing cluster. The jobs were run using 10 CPUs and 15 GB of memory. We used 23 hours as a wall time for the spatial transcriptomics experiments; on average, analyzing one slice using cEBMF takes about 3 hours. For the sparsity-driven covariate simulations and the tiled-clustering model simulations, we used the same computational resources but let the simulation run for 72 h.

# D ADDITIONAL FIGURES

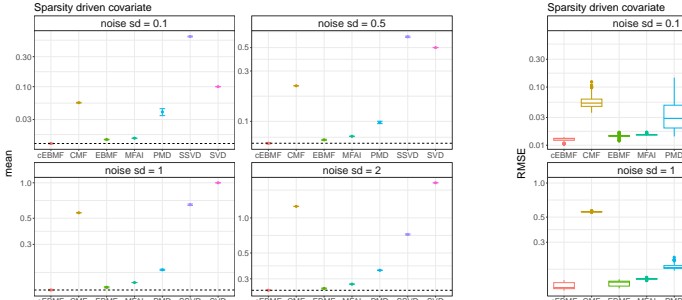

Figure 5: Comparative analysis of matrix factorization under various noise levels for the sparsity driven covariate simulations. The panel on the left-hand side displays the mean RMSE and their corresponding 95% confidence interval for each method. The y-axis represents the root mean squared error (RMSE). The y-axis is in the log scale for visualization purposes. The panel on the right-hand side displays the boxplot of the simulation results. The noise standard deviation for each simulation scenario is displayed in the facet title. The confidence intervals are computed assuming Normally distributed errors using standard closed-form formula.

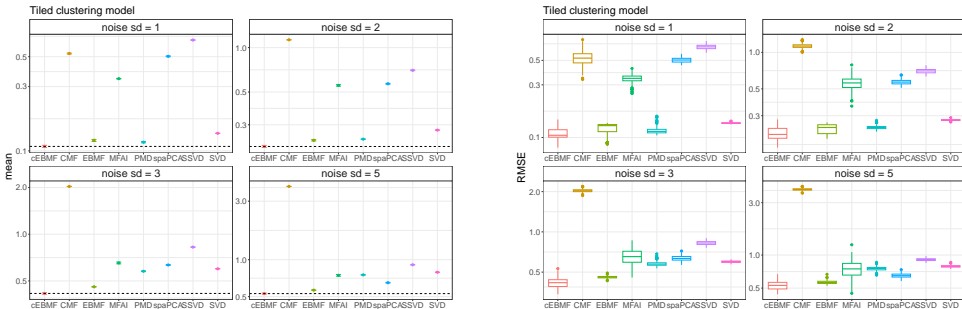

Figure 6: Comparative analysis of matrix factorization under various noise levels for the tiled clustering simulations. The panel on the left-hand side displays the mean RMSE and their corresponding 95% confidence interval for each method. The y-axis represents the root mean squared error (RMSE). The y-axis is in the log scale for visualization purposes. The panel on the right-hand side displays the boxplot of the simulation results. The noise standard deviation for each simulation scenario is displayed in the facet title. The confidence intervals are computed assuming Normally distributed errors using standard closed-form formula.

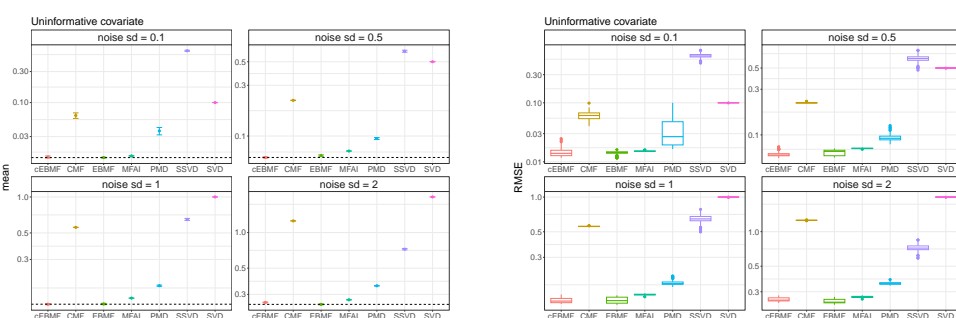

Figure 7: Comparative analysis of matrix factorization under various noise levels when the side information is not informative of the prior distribution. The panel on the left-hand side displays the mean RMSE and their corresponding 95% confidence interval for each method. The y-axis represents the root mean squared error (RMSE). The y-axis is in the log scale for visualization purposes. The panel on the right-hand side displays the boxplot of the simulation results. The noise standard deviation for each simulation scenario is displayed in the facet title. The confidence intervals are computed assuming Normally distributed errors using standard closed-form formula.

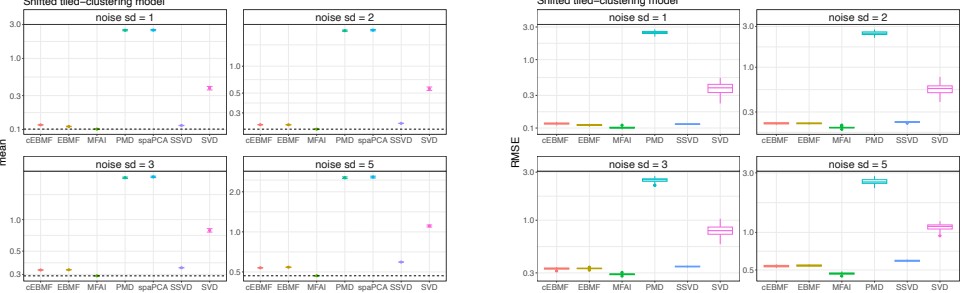

Figure 8: Comparative analysis of matrix factorization under various noise levels for the shifted tiled clustering simulations. The panel on the left-hand side displays the mean RMSE and their corresponding 95% confidence interval for each method. The y-axis represents the root mean squared error (RMSE). The y-axis is in the log scale for visualization purposes. The panel on the right-hand side displays the boxplot of the simulation results. The noise standard deviation for each simulation scenario is displayed in the facet title. The confidence intervals are computed assuming Normally distributed errors using standard closed-form formula.

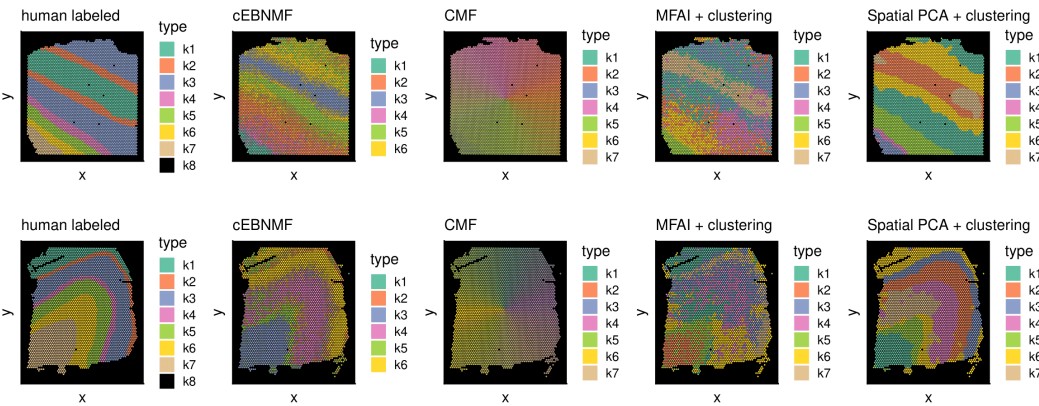

Figure 9: Results on slides 4 (A) and 10 (B) of the DLPFC spatial transcriptomics data (Pardo et al., 2022) as in the main text. In the NMF, EBMF and cEBMF results, each pixel $i$ is shown as a pie chart using the relative values of $i$-th row of $\mathbf{L}$ (after performing an "LDA-style" post-processing of $\mathbf{L}, \mathbf{F}$; Townes and Engelhardt 2023). Note that we only ran CMF and MFAi in this two examples, because CMF results are not informative and because MFAI is particularly long to run on this data.

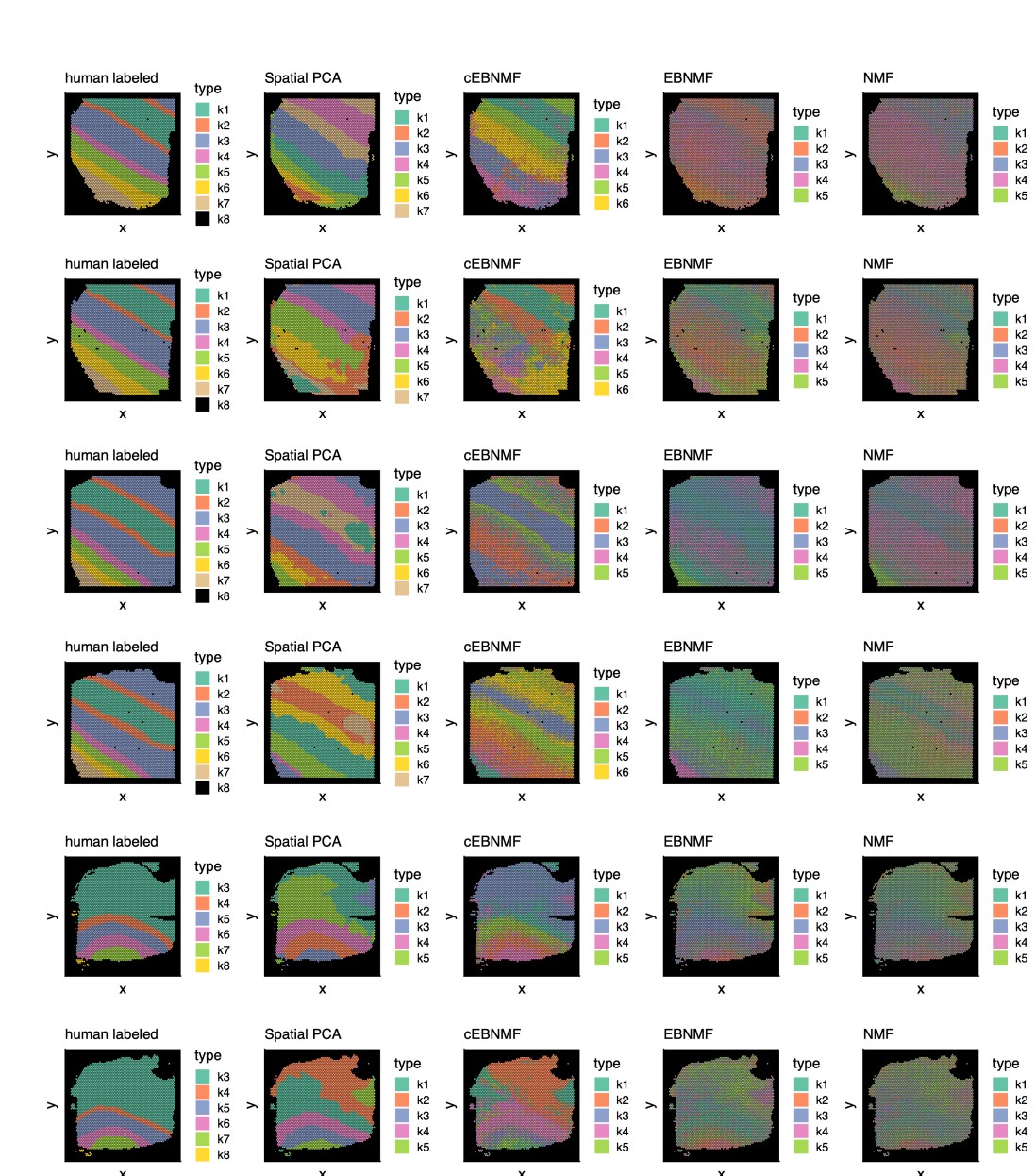

Figure 10: Slices 1 (top row) through 6 (bottom row).

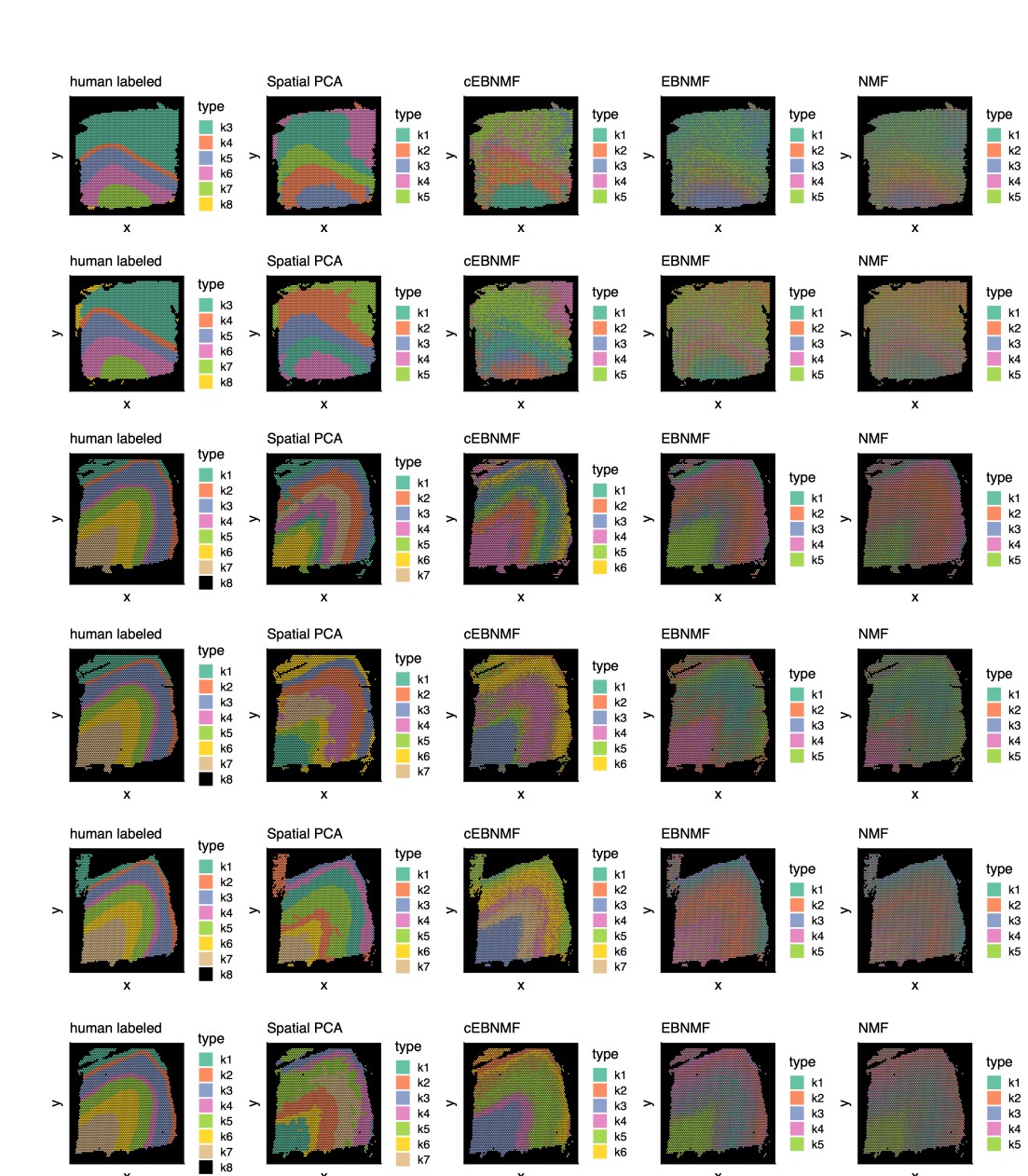

Figure 11: Slices 7 (top row) through 12 (bottom row).

