# OpenReview forum: "Covariate-moderated Empirical Bayes Matrix Factorization"
_ICLR.cc/2025/Conference — Submitted to ICLR 2025_

### Official Review · Reviewer_wcpG · 2024-11-01

**Soundness:** 3
**Presentation:** 3
**Contribution:** 2
**Rating:** 5
**Confidence:** 3

**Summary:**

This paper proposes a novel approach, covariate-moderated Empirical Bayes Matrix Factorization (cEBMF), which is aimed at enhancing the prediction of missing values in matrix factorization problems where side information can be leveraged flexibly. Compared to existing models, cEBMF allows prior knowledge to be incorporated into its design as a modular component, similar to a neural network module. The parameter estimation in this framework is based on an Empirical Bayes approach, and an estimation algorithm is derived accordingly. Experimental results on three datasets demonstrate that the proposed method achieves better performance.

**Strengths:**

- The cEBMF model introduces a novel methodology.
- The paper is well-organized and written clearly, with a structured overview of related work that positions the proposed method effectively.
- The code is provided, supporting reproducibility.

**Weaknesses:**

- There is a lack of comparison with neural network-based approaches. While several models capable of handling side information have been proposed in the literature [1-3], no comparison with these methods is provided.
- The primary contribution of this paper lies in its novel approach to incorporating prior knowledge. While this is valuable within the Bayesian community, its broader impact on the ICLR community, which tends to be more focused on neural network approaches, might not be as apparent.



1. Guo, H., Tang, R., Ye, Y., Li, Z., & He, X. (2017). DeepFM: A factorization-machine based neural network for CTR prediction. *Proceedings of the 26th International Joint Conference on Artificial Intelligence (IJCAI)*, 1725–1731. https://doi.org/10.24963/ijcai.2017/239

2. Cheng, H.-T., Koc, L., Harmsen, J., Shaked, T., Chandra, T., Aradhye, H., ... & Anil, R. (2016). Wide & deep learning for recommender systems. *Proceedings of the 1st Workshop on Deep Learning for Recommender Systems*, 7–10. https://doi.org/10.1145/2988450.2988454

3. Xiao, J., Ye, H., He, X., Zhang, H., Wu, F., & Chua, T.-S. (2017). Attentional factorization machines: Learning the weight of feature interactions via attention networks. *Proceedings of the 26th ACM International Conference on Information and Knowledge Management (CIKM '17)*, 1019–1028. https://doi.org/10.1145/3132847.3132953

**Questions:**

- Can the insights gained from this paper be applied to neural network modeling or other machine learning frameworks rather than the Bayesian framework? If so, could you elaborate on potential methods for achieving this?

---

> ### Author Response · Authors · 2024-11-18
>
> Thank you for your time and effort in reviewing our work. We greatly appreciate the positive comments, and the questions.
>
> Your questions were about neural-network-based matrix factorization approaches, and whether our approach would have benefits compared to these existing approaches.
>
> First of all, thank you for pointing us to these papers, some of which we were aware of already, but some are new to us. We could have done a better job in Sec. 2 ("Related work") citing neural-network-based matrix factorization approaches, and briefly explaining how they relate to the cEBMF framework we have proposed here. We will address this gap in the revised manuscript.
>
> To give some context, our paper was focussed on using matrix factorization methods to gain insight into the data. For example, in the spatial transcriptomics application, we are interested in learning about the biological processes (co-ordinated networks of genes) that give rise to the data. In the introduction, we wrote, "Matrix factorization methods, which include principal component analysis (PCA), factor analysis, and non-negative matrix factorization (NMF), are very widely used methods for inferring latent structure from data, performing exploratory data analyses and visualizing large data sets… When factorizing a matrix, say Z, the matrix may be
> accompanied with additional row or column data—side information—that may be able to guide the matrix factorization algorithm toward a more accurate or interpretable factorization." The "interpretable" part is important: in particular, NMF are known to yield interpretable ("parts-based") decompositions, which is why NMF is of special interest in many applications. By contrast, neural-network-based approaches can often yield very good predictive performance, but they suffer from poor interpretability. Therefore, we believe that there is room for different matrix factorization approaches as these approaches have different tradeoffs.
>
> While we agree that deep learning has become the leading topic in machine learning conferences, if you go to the landing page for ICLR (https://iclr.cc), the list of relevant topics includes many other topics beyond deep learning. We would therefore hope that you will consider our work as being a topic of interest to the ICLR community. We also invite you to review our responses to the other reviewers' comments, which we hope will give some more context to the contributions of our work.
>
> Finally we would like to highlight that matrix factorization methods now are used to "study" neural net embedding and our work could also be useful in this setting. For example, we are aware of several work in which researchers aim at reducing bias/toxicity of LLM using similar method.
>
> This research often assumes that
>
> embedding vector ≈ high-frequency vector + toxic vector + context-dependent vector
>
> And essentially use an SVD type of method for cornering toxic/bais subspace of the embedding vector. While these works are empirical this assumption seems to work well in practice. Hence having refined matrix factorization may help refine LLM alignment. In particular cEBMF could be use leverage network specific information to estimate the embedding decomposition.
>
> See the reference hereafter (we are not part of these works).
>
>
>
> - Andrew Lee, Xiaoyan Bai, Itamar Pres, Martin Wattenberg, Jonathan K Kummerfeld, and Rada
> Mihalcea. A mechanistic understanding of alignment algorithms: A case study on dpo and toxicity.
> arXiv preprint arXiv:2401.01967, 2024
> - Mor Geva, Avi Caciularu, Kevin Ro Wang, and Yoav Goldberg. Transformer feed-forward layers
> build predictions by promoting concepts in the vocabulary space. arXiv preprint arXiv:2203.14680,
> 2022
> - Kevin Meng, David Bau, Alex Andonian, and Yonatan Belinkov. Locating and editing factual
> associations in gpt. Advances in Neural Information Processing Systems,
> - Mor Geva, Roei Schuster, Jonathan Berant, and Omer Levy. Transformer feed-forward layers are
> key-value memories. In Proceedings of the 2021 Conference on Empirical Methods in Natural
> Language Processing, pp. 5484–5495, 2021
> Don't forget about pronouns: Removing gender bias in
> language models without losing factual gender information. In Proceedings of the 4th Workshop
> on Gender Bias in Natural Language Processing (GeBNLP)

---

> > ### Comment · Reviewer_wcpG · 2024-11-26
> >
> > Thank you for your response. I understand the importance of interpretability (I used to work in that field). However, I couldn't find evidence that cEBMF is suitable for such an application. The results in Fig 4 would be close to that line, but there's no quantitative evaluation. Is it possible to quantitatively measure that cEBMF is better in terms of interpretability?
> >
> > I appreciate the introduction of the applications for NNs. However, I still have the concern above, and I'll maintain the score.

---

### Official Review · Reviewer_3yAr · 2024-11-04

**Soundness:** 2
**Presentation:** 3
**Contribution:** 3
**Rating:** 5
**Confidence:** 5

**Summary:**

This paper proposes an extension of empirical Bayes matrix factorization (EBMF), by incorporating "side information" through an adaptive prior, resulting in a method called covariate-moderated EBMF, or cEBMF. One type of side information discussed in this paper is spatial information, motivated by the recent popular technology, spatial transcriptomics.

**Strengths:**

1. Matrix factorization is important to the community.

2. Spatial transcriptomics is an interesting application.

3. The cEBMF framework is broader than specific models, such as spatial models, or sparse models.

**Weaknesses:**

1. It seems that Section 3.2 is incomplete. My understanding is 3.2 discusses the general framework of cEBMF model, followed by some specific examples of side information. Then 3.2.1, as expected, discussed the case where the side information is factor sparsity. As a reader, I was expecting 3.2.2 to be another example, say when side information is spatial, as discussed in both the abstract and the introduction. However, there is no such an subsubsection, and actually 3.2.1 is the only subsubsection of 3.2. Did I miss anything here?

2. Following the above comment, in 4.1, the simulation settings include sparsity-driven covariate, and tiled-clustering model. However, there is no such a discussion on tiled-clustering model in 3.2. This setting seems a spatial example, so I think it makes more sense to have a section 3.2.2 for it.

3. The ST analysis need to be improved:

3.1 There is no quantitative metrics comparing methods, for example, Adjusted Rand Index, among others.

3.2 There are more competitors to be included, like those discussed in the introduction. One obvious example is NSF by Townes and Engelhardt.

3.3 Further results on other slides in Maynard dataset, shown in Figure 10 and 11, don't seem to support the claim that "cEBNMF tended to produce the largest improvements in accuracy". I might be wrong since this is purely visualization, which again requires some more quantitative scores to better compare the performance.


Minor issues:

Line 112-113, the citation Gopalan et al. (2014) should be in (), say \citep instead of \cite.

The order of methods in Figure 4, 9, 10, 11 are different.

In ST, "slide(s)" are used more frequently than "slice(s)" in my understanding. I personally prefer "slides", but I don't have any strong opinion on this. However, at least it should be consistent within this manuscript. A simple search would find 10 "slices", and two "slides".

**Questions:**

These are somewhat repetitive.

1. Can the authors extend 3.2, by adding more settings under which side information can be used (and how).

2. Since one major selling point of cEBMF is the flexibility in incorporating side information, is there any other examples in addition to sparsity and spatial information? If that's all, one potential argument why can't I just use sparse methods and/or spatial methods?

3. Can the authors improve the ST analysis in section 4.2? I mean to include more competitors and quantitative scores. Once the scores are included, it is easier for the audience to compare the methods, over all slides in the Maynard dataset.

---

> ### Author Response · Authors · 2024-11-18
>
> Thank you for your interest in our work and thoughtful comments. We appreciate your feedback, particularly on the spatial transcriptomics experiments,which we believe are interesting, but difficult to present to a machine learning audience. Thank you also pointing out the minor mistakes in our paper—we will correct these in the revised manuscript.   Below, we address your comments.
>
>
> - You wrote, *"It seems that Section 3.2 is incomplete. My understanding is 3.2 discusses the general framework of cEBMF model, followed by some specific examples of side information. Then 3.2.1, as expected, discussed the case where the side information is factor sparsity. As a reader, I was expecting 3.2.2 to be another example, say when side information is spatial, as discussed in both the abstract and the introduction. However, there is no such subsubsection, and actually 3.2.1 is the only subsubsection of 3.2. Did I miss anything here?"*
>
> Due to ICLR’s page limits, we could not elaborate on the priors used in detail. However, we provided the EBNMF and cEBNMF model details for spatial transcriptomics: EBNMF uses point-exponential priors, while cEBNMF employs scale mixtures of exponential priors, as in other examples (see "tiled-clustering model" on p. 7). Further details are in the Appendix (Sec. C.5, "Spatial transcriptomics experiments").
>
> - You wrote, *"Since one major selling point of cEBMF is the flexibility in incorporating side information, is there any other examples in addition to sparsity and spatial information? If that's all, one potential argument why can't I just use sparse methods and/or spatial methods?"*
>
> Side information has diverse applications, including images, text, and graphs, where specific approaches have been developed. We aim to unify these within a single framework. For example, we mentioned using histological images in Sec. 5 ("Limitations"), although the poor quality of spatialLIBD images limited their utility. Another example is collaborative filtering (Sec. 4.2), where movie genres informed the factorization.
>
> Using side information to enforce sparsity is itself a broad topic, and cEBMF allows for exploring various sparsity-driving priors, offering flexibility beyond traditional sparse or spatial methods.
> You also commented on the spatial transcriptomics experiment:
> 1. Lack of quantitative metrics for comparison, e.g., Adjusted Rand Index (ARI).
> 2.  Missing competitors, e.g., NSF by Townes and Engelhardt.
> 3.  Results on other slides in Maynard dataset (Figures 10–11) don’t clearly support the claim that *"cEBNMF tended to produce the largest improvements in accuracy."*
>
>
> -  Our goal was not to recover human labeling but to generate biologically meaningful representations (e.g., cell types, tissue regions). As noted, "Our aim was to generate a parts-based representation of the data…" Human labeling serves as a reference, not ground truth, as experts cannot fully determine the underlying biological processes. Thus, quantitative metrics like ARI are less meaningful here. By contrast, MovieLens offers ground truth, which is why we included quantitative comparisons for that experiment.
> - We agree NSF is a notable competitor. Spatial transcriptomics was an example application, not the paper’s primary focus, so we didn’t include all methods. Spatial PCA was prioritized due to its citation frequency and well-developed software. However, we are now running NSF and expect to include results in the revised manuscript.
> - The claim about accuracy improvements applies to MovieLens, where we wrote, "Both MFAI and cEBNMF were able to use the side information (the movie genres) to provide improvements over EBNMF, and cEBNMF tended to produce the largest improvements in accuracy. *" For spatial transcriptomics, we emphasized that "Comparatively, the cEBNMF results in slices 4 and 9 capture the expert labeling much more closely…"* Spatial PCA performed poorly on slice 11 and differed significantly from human-labeled data on slice 10.
>
>
> Finally, spatial transcriptomics deserved more than one page, but space constraints limited us. A future study will focus on spatial transcriptomics, including broader comparisons and validation with more datasets.
> The reviewer can access the high resolution images of the plot for slice 10 and 11 in the submitted material at this location in the zip folder \ICLR_submission\cEBMF_RCC_experiments\plot .
>
> We also invite you to review our responses to the other reviewers' comments, which we hope will give some more context to the contributions of our work.

---

> > ### Comment · Reviewer_3yAr · 2024-11-24
> >
> > Thanks for the responses.
> >
> > Regarding section 3.2. I understand the page limit, and I agree that it's ok to discuss other methods in the appendix. However, in such case, I wouldn't use subsubsection since there 3.2.1 is the only one inside 3.2
> >
> > Regarding the ST experiment, I am still not convinced by the rebuttal for the following reasons:
> >
> > 1. The authors argue that the expert label is not ground truth, but they use the label to support their results "Comparatively, the cEBNMF results in slices 4 and 9 capture the expert labeling much more closely, with most factors showing a clear spatial quality".
> >
> > 2. Again, spatial PCA is a very simple, arguably the most simple method here. It's ok as a baseline, but more comparisons are expected.
> >
> > 3. Overall, this application is not convincing enough to show the superior performance of the proposed method.
> >
> > As a result, I decide to keep my score as for now.

---

### Official Review · Reviewer_5MJg · 2024-11-04

**Soundness:** 3
**Presentation:** 4
**Contribution:** 3
**Rating:** 8
**Confidence:** 4

**Summary:**

The paper proposed a new matrix factorization method with covariates for rows/columns included in the model. It generalizes previous studies on nonnegative matrix factorization and empirical Bayes matrix factorization. The proposed model can utilize any side information that can be included in a probabilistic model with the flexibility of little to no assumption on the factors.

**Strengths:**

1. Treating side information through empirical Bayes in the matrix factorization problem is novel.
2. The paper provides clear derivations and algorithms for the proposed methodology. Part of the result is justified in the appendix.

**Weaknesses:**

My major concern is the parametric prior assumption in Eq. (4). It can easily overfit the model given the flexibility in Eq. (4). Although the authors mentioned the overfitting problem in the limitation section, it cannot be overlooked.  Details are provided in the Questions.

**Questions:**

1. The parametric assumption in Eq. (4)  is confusing --- it can easily lead to an overfitted model if the index of the distribution family is arbitrarily related to $\mathbf{x}\_i$. More constraints should be given to control the model complexity.
For example, it could be $\ell_{ij}\sim g_k^{(l)}(q_i)\in \mathcal G_{l,k}$ with $q_i=q_i(\textbf{x}_i)\in\mathcal Q$, where restriction on the function family $\mathcal Q$ helps control the complexity. Actually, the authors are doing this in later illustrations. For example, in the spike-and-slab prior example in Sec. 3.2.1, $q_i$ is a logistic function of $\mathbf{x}_i$. Further discussion on the choices of the parametric families and the constraints on controlling the model complexity should be provided.
2. Does the proposed model assume a known number of factors $K$? If not, how should $K$ be determined?
3. If $\mathbf{Z}$ is a symmetric matrix (e.g. covariance) such that $\mathbf{L}=\mathbf{F}$, is there any change to the current process?
4. For the example in Sec 4.2, the side information $\mathbf{X}$ is the genre of the movies. If every movie belongs to exactly one genre (correct me if I was wrong), then the empirical prior appears to be a hierarchical prior (19 priors with parameters from a common hyperprior). Then I don't see why the proposed model is needed.
5. For the example in Sec 4.2, what is the side information for columns, i.e. $\mathbf{Y}$?
6. Other minor typo/writing issues:
    1. In Eq. (3), $\ell_{ik}$ and $f_{jk}$ are not defined.
    2. In Eq. (5), $\mathbf{\omega}$ is not defined.
    3. In Eqs. (14) and (16), should Eq. (14) be $\overline{\mathbf{R}}^k = \overline{\mathbf{R}} + \overline{\mathbf{\ell}}_k\overline{\mathbf{f}}_k^T$ and Eq. (16) be the other way?

---

> ### Author Response · Authors · 2024-11-18
>
> Thank you for your interest in our work, and for the positive and constructive feedback.
>
> We will attempt to address your comments and questions.
>
> - First, you wrote, *"My major concern is the parametric prior assumption in Eq. (4). It can easily overfit the model given the flexibility in Eq. (4). Although the authors mentioned the overfitting problem in the limitation section, it cannot be overlooked."*
>
> And relatedly, in Question 1:* "The parametric assumption in Eq. (4) is confusing—it can easily lead to an overfitted model if the index of the distribution family is arbitrarily related to x_i. More constraints should be given to control the model complexity…"*
>
> We agree that overfitting is potentially a problem, but in machine learning there are many well-developed strategies for dealing with overfitting—some of the simplest and most widely used being penalty approaches and cross-validation—so we feel that this is a limitation that can be overcome without too much effort.
>
> The related question (Question 1) is also about overfitting, but this is actually a strength of the EBMF and cEBMF frameworks: dealing with overfitting is baked in to the framework by using priors that control the complexity of the model (by contrast, standard approaches such as PCA and NMF do not deal with overfitting). Of course, there are different priors one could use—and there is a large literature on this topic—but in the paper we proposed a couple of different priors (such as the spike-and-slab) that we think could be generally useful.
>
> Regarding your comment about the parametric assumptions on prior distribution families, in the presentation we assumed parametric priors. But this was mainly made for ease of presentation. It is more intuitive to think about incorporating side information into parametric priors, e.g., using a logistic regression to parameterize the weights in the "spike-and-slab" prior (eq. 6). But in fact the cEBMF is more general than this. Consider that, in cEBMF, the model is fitted by iteratively solving a series of simpler problems, which we call "covariate-moderated Empirical Bayes normal means" (cEBNM) problems. Any prior family would be admissible under the cEBMF framework as long as the cEBNM mapping (eq. 13) is computable.
>
> - Question 2: Does the proposed model assume a known number of factors K? If not, how should be determined?
>
> Yes, so one advantage of the empirical Bayes approach is that it can automatically choose K, at least under certain conditions. (It requires the model to be correct—model misspecification can lead to an incorrect setting of K.) We describe how to automate choosing K in the supplement. Intuitively, if the prior includes a point-mass at zero, the model will stop adding factors when it encounters a "zero factor".
>
> - Question 3: If Z is a symmetric matrix (e.g., covariance) such that L = F, is there any change to the current process?
>
> Great question. We have not yet considered the symmetric case, but we are interested in it, and we believe our approach could be extended to handle this case. There are at least two possible approaches. the first is to simply apply the method as is, without enforcing L = F. In practice, because Z is symmetric, the estimated L and F will often be similar to one another. This approach, despite seeming to be ad hoc, actually works quite well, as has been shown previously (A. Chaturvedi and J. D. Carroll. An alternating combinatorial optimization approach to fitting the INDCLUS and generalized INDCLUS models. Journal of Classification,  1994., Zhihui Zhu, Xiao Li, Kai Liu, and Qiuwei Li. Dropping Symmetry for Fast Symmetric Nonnegative Matrix Factorization. NeurIPS, 2018.)  Another approach could be to factorize the Cholesky factors of Z.
>
> - Question 4: For the example in Sec 4.2, the side information X is the genre of the movies. If every movie belongs to exactly one genre (correct me if I was wrong), then the empirical prior appears to be a hierarchical prior (19 priors with parameters from a common hyperprior). Then I don't see why the proposed model is needed.
>
> In Sec. 4.2, the movies belong to different genres and some are combinations of genres (e.g., a movie can be both a drama and a comedy).
>
> - Question 5: For the example in Sec 4.2, what is the side information for columns (i.e., Y)?
>
> In the MovieLens data, Y is user information (genre, age), but we did not use this information to keep the comparison fair with MFAI.
>
> -Regarding your minor comments (Question 6), we agree that points 1 and 2 need to be clarified, and we will do so in the revised manuscript. Regarding point 3, eqs. 14 and 16 indeed have the incorrect signs—thank you for spotting this mistake. This will also be fixed in the revised paper.
>
> We also invite you to review our responses to the other reviewers' comments, which we hope will give some more context to the contributions of our work.

---

> > ### Comment · Reviewer_5MJg · 2024-11-25
> >
> > Thank you for the comments and clarifications. I decide to keep my score.

---

### Official Review · Reviewer_JVaU · 2024-11-04

**Soundness:** 3
**Presentation:** 3
**Contribution:** 2
**Rating:** 6
**Confidence:** 2

**Summary:**

The paper proposes a modular framework for empirical Bayes matrix factorization that can leverage a large variety of models, and can use families of priors that are flexible in form to accommodate different assumptions and constraints, and allows automatic selection of the hyperparameters. Various experiments are conducted to illustrate the effectiveness.

**Strengths:**

The proposed framework is modular, flexible, and general, encompassing many previous studies as special cases. The presentation of the core ideas and algorithms is clear, supported by the appendix. The experimental results sufficiently demonstrate the method's effectiveness. The paper provides a well-balanced discussion of the approach's strengths and limitations.

**Weaknesses:**

The overall technical contribution appears moderate, as the generalization from MFAI seems straightforward. The key challenges in model formulation and algorithm design could be better articulated.

**Questions:**

It would be helpful if the authors could elaborate on the specific obstacles encountered in extending MFAI.

---

> ### Author Response · Authors · 2024-11-18
>
> Thank you for your interest in our work, and for the positive and constructive feedback.
>
> We would like to respond to this comment: *"The overall technical contribution appears moderate, as the generalization from MFAI seems straightforward. The key challenges in model formulation and algorithm design could be better articulated."*
>
> We developed cEBMF independently of MFAI and only became aware of this work recently. Nonetheless, we would like to emphasize some key differences between MFAI and cEBMF. In Sec. 2, we wrote: *"MFAI is in fact a special case of cEBMF in which the priors on F are normal and the prior means are informed by the covariates. Similar to cEBMF, MFAI allows these priors to be adapted separately for each factor k. However, it is not nearly as general as cEBMF; it implements only a single model, a single prior family of a specific parametric form, a specific procedure for fitting these priors, and it only accommodates row-wise side information."* Our empirical comparisons with MFAI underscore the importance of having a flexible matrix factorization framework in which the prior assumptions can be tailored to the target data set. Also, because the cEBMF framework is very flexible, it can sometimes accommodate the specific modeling choices made by other methods.
>
> Regarding your comment that the generalization from MFAI seems "straightforward", it one sense we agree with you: it is straightforward to write down on paper a general framework for covariate-moderated matrix factorization. But what makes our approach particularly attractive is that it is modular; that is, it is easy to extend cEBMF by "plugging in" different cEBNM solvers (eq. 13), then reusing the cEBMF algorithm (Sec. 3.3.2). Although this may seem straightforward in retrospect, realizing such a framework builds on several recent technical contributions and software, and to our knowledge we are the first to show that this sort of modular framework is possible, and that it works well with different types of data and side information.
>
> We also invite you to review our responses to the other reviewers' comments, which we hope will give some more context to the contributions of our work.

---

### Official Review · Reviewer_ouMG · 2024-11-05

**Soundness:** 3
**Presentation:** 3
**Contribution:** 2
**Rating:** 5
**Confidence:** 4

**Summary:**

The paper proposes a covariate-moderated empirical Bayes matrix factorization, which borrows information from the side information. The side information is modeled as a prior information. Comparisons are made to some competitors through simulations and real data analysis.

**Strengths:**

Paper is well-written with detailed introduction on empirical bayes and clarifications on details.

The whole paper is easy to read and technical parts are easy to follow.

**Weaknesses:**

All the empirical studies are simple. The dimensions of the simulated data are not large. 1000*200 matrix Z is not enough to show the effectiveness. Even the real datasets are relatively small.

**Questions:**

The side information can be treated as other views for learning, which is multi-view learning. There are tons of papers about this. It can also be treated as source data in transfer learning. Authors need to discuss why the side information is modeled into the prior as a preferred way.

Authors mentioned on Line 183 that it can leverage many models. Can authors tell more about the details? Do authors mean that the prior information can be found via modeling?

How is the rank determined in the paper? It is very difficult to identify it. Also, in the empirical studies, K is treated as known in the simulation. In MovieLens, how all methods choose K based on complexity of data (Line 469)? In DLPFC, I see most people use K=15. If doing clustering, authors need to talk about choosing the number of clusters.....

In Figure 2, if I am right, the uninformative covariate should have worse results than sparsity-driven covariate, but they seem to have better results. Shifted tiled-clustering should have worse results, but MFAI got better. Can authors clarify these?

I used the DLPFC dataset multiple times and it may not be a good way to present it here. Literature about spatial transcriptomics focuses a lot on incorporating the spatial information, e.g., the empirical Bayes model for spatial transcriptomics. How is the spatial information used in this method? People typically use it for clustering, so authors may show the clustering results, ARI etc. Many methods haven been demonstrate on this dataset as well.

---

> ### Author Response · Authors · 2024-11-18
>
> Thank you for reading our paper and providing thoughtful feedback. Below, we address your comments and questions.
>
>
>
> - You wrote, *"The side information can be treated as other views for learning, which is multiview learning. There are tons of papers about this. It can also be treated as source data in transfer learning. Authors need to discuss why the side information is modeled into the prior as a preferred way."*
>
> We are aware of the extensive multiview learning literature. Our focus is on a hierarchical Bayes approach to integrate dependencies between side information and matrix factorization in a data-driven manner. Multiview learning tends to focus on modeling commonalities between more than the dependencies. We use a special case of multiview learning in our comparison (collaborative matrix factorization), which assumes that, to a certain extent, the factorization of the side information is similar to the data of interest. In the case that the side information is spatial (rank 2), it is simply not possible to find a shared factorization with the matrix of interest if its rank is >2. On the other hand, multiview learning approaches typically make assumptions that aren't suitable for some applications, such as spatial transcriptomics data.
>
> - You wrote, *"Authors mentioned on Line 183 that it can leverage many models. Can authors tell more about the details? Do authors mean that the prior information can be found via modeling?"*
>
> Yes, the learning algorithm of cEBMF is agnostic to the type of neural network used (e.g., GNN, transformer), provided it is fitted using the loss in Eq. 11 (the marginal likelihood of the EBNM model). For instance, image data, such as cell images, could be used to inform the prior via a CNN.
>
> - You wrote, *"How is the rank determined in the paper? It is very difficult to identify it. Also, in the empirical studies, K is treated as known in the simulation. In MovieLens, how all methods choose K based on complexity of data (Line 469)? In DLPFC, I see most people use K=15. If doing clustering, authors need to talk about choosing the number of clusters…"*
>
> One advantage of the empirical Bayes approach is that it can automatically choose K, at least under certain conditions. We describe how to automate choosing K in the supplement. Intuitively, if the prior includes a point-mass at zero, the model will stop adding factors when it encounters a "zero factor".
>
> - You wrote,*"In Figure 2, if I am right, the uninformative covariate should have worse results than sparsity-driven covariate, but they seem to have better results. Shifted tiled-clustering should have worse results, but MFAI got better. Can authors clarify these?"*
> We view the uninformative covariate simulation as a "sanity check" to show that cEBMF performs similarly to EBMF in this setting.
>
> The shifted tiled-clustering experiment was specifically designed to be a more challenging example where the true hierarchical model cannot be recovered by the parameterization of the prior we used. The intent was to assess robustness to model misspecification. Since this simulation matches the MFAI modeling assumption, it makes sense that MFAI performed better than cEBMF in this case.
>
> - You wrote, *"I used the DLPFC dataset multiple times, and it may not be a good way to present it here. Literature about spatial transcriptomics focuses a lot on incorporating the spatial information,[ …] . How is the spatial information used in this method? People typically use it for clustering, so authors may show the clustering results, ARI etc."*
>
> The spatial information (2-d pixel locations) informs the prior on the L matrix, as illustrated in Fig. 1. While clustering is common for spatial transcriptomics (e.g., in Spatial PCA), it captures limited structure and often misses biologically meaningful patterns. Clustering assumes orthogonal factors, while NMF can reveal continuous variation and mixtures of cell types. As we noted, "NMF methods can capture continuous variation in expression within and across cell types or regions… whereas clustering cannot." This broader scope makes matrix factorization especially useful for spatial transcriptomics.
>
> - You wrote, *"All the empirical studies are simple. The dimensions of the simulated data are not large. 1000x200 matrix Z is not enough to show the effectiveness. Even the real datasets are relatively small."*
>
> While cEBMF can handle large data sets (see computational complexity notes in Sec. 3), our experiments focused on comparing cEBMF to other matrix factorization methods. Many comparable methods (e.g., MFAI) scale poorly, limiting us to smaller data sets. For instance, MFAI was much slower than cEBMF on the DLPFC data and generally struggles with large single-cell data sets.
>
>
> We also invite you to review our responses to the other reviewers' comments. We hope these responses provide helpful context and underscore the contributions of our work. Thank you for your consideration.

---

> > ### Comment · Reviewer_ouMG · 2024-11-23
> >
> > I thank the authors for their responses. However, there are a few points unaddressed. For example, multiview learning has been used in spatial transcriptomics data a lot. That multi view learning is not suitable for ST data needs to discuss. Clustering is one of the fundamental tasks in unsupervised learning, plus that NMF is used frequently in clustering and DLPFC is a benchmark dataset in ST, demonstrating the capacity of clustering is another way to show the importance of work. Overall, I keep my score.

---

### Official Review · Reviewer_fTeh · 2024-11-05

**Soundness:** 2
**Presentation:** 2
**Contribution:** 2
**Rating:** 5
**Confidence:** 4

**Summary:**

The work deals with a Bayesian matrix factorization (BMF) framework that can accommodate the side information using a parametrized representation, e.g., neural network. Such side information allows the framework to handle any prior of the latent in flexible manner. The work extends the EBMF framework proposed in [Wang and Stephens, 2021; Willwersheid, 2021)] along with the classic normal means model. Experiments with both simulated data and real data are presented to support the claims.

**Strengths:**

Strengths:

1.	The wok is an interesting combination of connecting classical models with more expressive models in deep learning.

2.	Real data experiments especially the genomic data is insightful and the performance of the proposed approach looks reasonable.

**Weaknesses:**

Weakness:

1.	The discussions and sections could have been organized better. I think, many important technical details and discussions are moved supplementary, which makes hard to verify the technical soundness and the reasoning.

2.	Some relevant baselines are missing from discussion and empirical study.

**Questions:**

1.	In the introduction, a figure or a clear description could have helped what are the side information in the context of the described genomic data.

2.	It is commented that “Further, there are sometimes benefits to not making strong assumptions about the spatial organization of the data even when we know the data are spatial.”. This is a bit unclear statement as spatial PCA seems perform well in the real data experiments in Figure 4. Could you clarify this statement?

3.	Figure 1 is hard to understand the clustering performance of the different methods. Instead of PCs, it would have been more natural to compare the clustering of the original points in x-y domain. Also, how does different prior assumptions.

4.	What are the connections with deep matrix factorization frameworks? There exists some works by directly modeling the latent using the deep architectures with prior information, e.g.,
a.	Wang, Jianyu, and Xiao-Lei Zhang. "Deep NMF topic modeling." Neurocomputing 515 (2023): 157-173.
b.	Xue, Hong-Jian, et al. "Deep matrix factorization models for recommender systems." IJCAI. Vol. 17. 2017.
And the related references.
These types of frameworks are non-Bayesian, without making any distributional assumptions on the prior. I think, they are very related to the proposed method and could not find any discussion or empirical experiments in the paper.

5.	It is unclear how does the constraints are handled in this case, for e.g., nonnegative constraints in Eq. (15) in the algorithm design.

6.	Also, if there are missing side information (i.e., matrix entry of Z is present, but side information from X is missing), will the method be able to handle it?

7.	“In practice, the full posterior q is not needed; the first and second posterior moments are sufficient”. However, looking at Eq. (12), it seems that the full posterior is needed (notations are also confusing here, $p$ vs $q$). It is unclear how does this translate to the moments of  $\ell_{ik}$ and $f_{ik}$. Are there for a specific family of distributions?

8.	In the real data experiments with collaborative filtering, there exists side information for only columns. In that case, how does the algorithm handle it? In spatial transcriptomics data, no side information is specified. Spatial PCA seems performing well in the real data. What about the runtime performance of the proposed algorithm and how does it compared to the competing baselines?

9.	I feel the series of equations 31-36 has some issues. I doubt if (31) and (32) are equal, which then questions the correctness of the remaining equations. Also, the distribution $q_{\beta}$ is not defined properly. As the details are missing in the main paper, the soundness of algorithm is hard to verify. Please add more clarity to the algorithm design in the main text.

10.	Any insights about identifiability of this approach which is a key consideration in matrix factorization-based models?

11.	Minor comments:

a.	Some typos: Notation in (7). Notation confusion: side information is notated using $\bm x_i$ or $\bm y_i$, but then $\bm d_i$ is used in (9)

b.	“D is an invertible diagonal matrix” in Page 4. D can be any invertible matrix

---

> ### Author Response · Authors · 2024-11-18
>
> Thank you for your detailed comments and  interest on our paper. We hope you will reconsider its value in light of our responses below. Matrix factorization is an important topic in machine learning and we believe that this work will generate considerable interest and discussion at ICLR 2025.
>
> **Model and Algorithm:**
> You raised several points about the presentation of the model, algorithm, and missing definitions in the equations. While some details are already included, we will revise the manuscript to ensure absolute clarity. E.g., you wrote, "a figure or a clear description could have helped what are the side information in the context of the described genomic data," and "In spatial transcriptomics data, no side information is specified." We were explicit that the side information for genomic data is the 2-d pixel locations,  "A recent prominent example of this in genomics research is spatial transcriptomics data… the 2-d coordinates of the pixels also provide important information about the biological context of the cells." Additionally, Fig. 4 illustrates this.
>
> You also asked how cEBMF handles side information:
>
> "If there are missing side information [...], will the method be able to handle it?", "In the real data experiments with collaborative filtering, there exists side information for only columns. In that case, how does the algorithm handle it?"
>
> In Sec. 3.2, we wrote, "In covariate-moderated EBMF (cEBMF), we assume that we have some side information (covariates) for rows and/or columns of Z.". Note the "and/or", in short, we do not require side information for both rows and columns.
>
> **Handling Constraints:**
> You asked, "It is unclear how does the constraints are handled in this case, for e.g., nonnegative constraints in Eq. (15)." Constraints, such as non-negativity, are a key aspect of our method. We addressed this in Sec. 3.1, "The flexibility of EBMF comes from the wide range of different possible prior families. Different choices of prior family correspond to different existing matrix factorization methods… families that only contain distributions with non-negative support result in empirical Bayes versions of non-negative matrix factorization." Constraints are not applied in Eq. 15,  but rather in Eq. 17, where priors impose constraints. E.g., a prior with support only on non-negative values results in non-negative posterior distributions and estimates. We will clarify this further.
>
> **Mathematical Correctness:**
> You raised a few concerns about missing definitions and mathematical correctness, particularly questioning the equivalence of Eqs. (31) and (32). This result directly applies Bayes' Theorem, and we will expand the explanation accompanying Eqs. (31–35) to make the derivation more transparent.
>
> **Performance Comparisons:**
> Several of your questions (Questions 2–4 and 8) reflect a potential misunderstanding of our goals. While matrix factorization methods can serve diverse purposes, our focus is on gaining insights from data, not just predictive accuracy. E.g., in the spatial transcriptomics application, our goal is to uncover the biological processes (co-ordinated networks of genes) underlying the data. We wrote : "Matrix factorization methods, which include principal component analysis (PCA), factor analysis, and non-negative matrix factorization (NMF), are very widely used methods for inferring latent structure from data, [...] visualizing large data sets… When factorizing a matrix, say Z, the matrix may be accompanied with additional row or column data—side information—that may guide the matrix factorization algorithm toward a more accurate or interpretable factorization." The emphasis on "interpretable" is crucial. NMF is valued for producing interpretable "parts-based" decompositions, whereas neural-network-based methods often excel in prediction but lack interpretability, which is typically not emphasized in the referenced papers.
>
> We use predictive metrics,e.g., in collaborative filtering experiments for quantitative evaluation, our ultimate aim is interpretability. For spatial transcriptomics, the goal is "to generate a parts-based representation of the data, with the hopes that the parts would resolve to biologically meaningful units (e.g., cell types, tissue regions)." For comparison in Fig. 4, we included a human labeling, but note that this labeling is not ground truth, even experts cannot definitively know the underlying biological processes. While Spatial PCA seems to "perform well", it clusters data but cannot reflect mixtures of biological processes, which our approach captures. These mixtures, though sometimes appearing "messy," align with the complex reality of biology. Notably, cEBNMF was not expected to outperform Spatial PCA, a method specifically tailored to spatial transcriptomics data. Nonetheless, cEBNMF achieves comparable results in capturing human labeling and spatial organization, demonstrating its empirical validity as an assumption-free framework.

---

> > ### Comment · Reviewer_fTeh · 2024-11-20
> > **Response to Author's comments**
> >
> > I thank the authors for the responses to my questions. However, I am not fully convinced about some of the responses. For example, my question about handling missing side information was about how it is being technically handled in the algorithm in Sec. 3.3.2. But your response was "In Sec. 3.2, we wrote, "In covariate-moderated EBMF (cEBMF), we assume that we have some side information (covariates) for rows and/or columns of Z.". Note the "and/or", in short, we do not require side information for both rows and columns.". This does not answer the technical approach.
> >
> > Another aspect still not clear is about the identifiability question, the run time, and performance comparison with deep NMF models.
> > At this point. I have decided to keep my score.

---

> > > ### Author Response · Authors · 2024-11-21
> > >
> > > Dear reviewer,
> > >
> > >
> > >  Thank you for taking the time to engage with our work and for raising thoughtful questions. We appreciate the opportunity to clarify the points you mentioned further. Due to the character limit in our previous response, we regret that we could not fully elaborate on all your concerns, but we aim to address them here in detail.
> > >
> > >  * Handling Missing Side Information   :
> > >
> > > When there is no side information, Equation (11) in our paper reduces to:   $\mathcal{L}(\pi) = p(\hat{\beta} \mid s, \pi) = \prod_{i=1}^n \int \mathcal{N}(\hat{\beta}_i; \beta_i, s_i^2) g(\beta_i; \pi) \, d\beta_i, $
> > >
> > > for example we set $g(\cdot; \pi)$ to be a mixture of a Gaussian and a point mass in tiled clustering simulation scenario. . Specifically:
> > >
> > >
> > >
> > > $ g(\cdot; \pi) = \pi_0 \delta_0(\cdot) + \sum_{k=1}^K \pi_k \mathcal{N}(\cdot; 0, \sigma_k^2). $
> > >
> > >
> > >  In this case, the problem simplifies to maximizing the following marginal likelihood with respect to the mixture proportions \(\pi\):   $\mathcal{L}(\pi) = \prod_{i=1}^n \left[ \sum_{k=0}^K \pi_k \mathcal{N}(\hat{\beta}_i; 0, \sigma_k^2 + s_i^2) \right].$ This optimization can be efficiently performed using standard methods such as the Expectation-Maximization (EM) algorithm or Sequential Quadratic Programming (SQP). In practice, we employ the fast algorithm described in the paper by Youngseok Kim et al. (2020), \textit{A Fast Algorithm for Maximum Likelihood Estimation of Mixture Proportions Using Sequential Quadratic Programming} (Journal of Computational and Graphical Statistics). We hope this clarifies the technical details of handling missing side information in our approach.
> > >
> > >
> > >  * Identifiability
> > >
> > > We agree that identifiability is an important consideration in matrix factorization models. As is well-known, matrix factorizations are generally not identifiable without additional constraints. In our context, identifiability can be addressed by applying a row and column permutation of the inferred matrices, which is a standard strategy in the field. For further details, we refer to works such as:     DOI:  10.1214/24-BA1423   ,   DOI:  10.1137/22M1507553
> > >  However, we did not explicitly discuss this topic in our manuscript, as it is generally treated as a separate topic from the modeling itself. We appreciate your bringing up this point and would be happy to expand on it in a future version of the paper.
> > >
> > > * Comparison with Deep NMF Models
> > >
> > >  Regarding performance comparison with deep NMF models: we compared our approach with Collaborative Matrix Factorization (CMF), which is a type of deep NMF model that incorporates side information. As shown in both our simulation studies and the application to spatial transcriptomics, our covariate-moderated EBMF (cEBMF) consistently outperforms CMF in these scenarios.
> > >
> > >  We hope that this additional clarification addresses your concerns. Thank you again for your detailed review,   we are happy to provide further explanations if needed.

---

### Author Response · Authors · 2024-11-18
**Global response**

We are grateful to receive 6 reviews for our submission, "Covariate-moderated Empirical Bayes Matrix Factorization". We appreciate that reviewing takes a lot time and effort, but we hope you will take a little bit more time to read and consider our responses to each of the reviews.

As this discussion has shown, matrix factorization remains a topic of wide interest in machine learning. We view our work as making an important contribution to this topic by proposing a flexible and easily extensible matrix factorization framework that allows for different modeling assumptions and constraints.

---

### Meta-Review · Area_Chair_vCBS · 2024-12-21

**Metareview:**

This paper proposes a method to use side information into the bayesian matrix factorization framework. This paper had interesting back and forth between the reviewers and the authors. At the end of the rebuttal period, there are multiple non-trivial concerns raised by the reviewers that were left unaddressed. The authors only partially responded to several clarifications requested by the reviewer fTeh, which should make the paper stronger. Similarly, further clarifications are needed about multiview learning and clustering. The need for more empirical results beyond simple baselines and for quantifying the interpretability was also emphasized to ascertain flexibility of the method.

**Additional Comments On Reviewer Discussion:**

Many of the points raised by the reviewer fTeh were left unanswered by the authors. Similarly at the end of the discussion period, some points raised by other reviewers on the need for non-trivial baselines, quantifying interpretability, and additional clarifications on multiview learning (which the authors claim to be a siginificant application for their method) are still lacking.

---

### Decision · Program_Chairs · 2025-01-22

Reject